# Oncolytic Virotherapy in Glioma Tumors

**DOI:** 10.3390/ijms21207604

**Published:** 2020-10-14

**Authors:** Sergio Rius-Rocabert, Noemí García-Romero, Antonia García, Angel Ayuso-Sacido, Estanislao Nistal-Villan

**Affiliations:** 1Microbiology Section, Departamento de Ciencias Farmacéuticas y de la Salud, Facultad de Farmacia, Universidad San Pablo-CEU, 28668 Madrid, Spain; ser.rius.ce@ceindo.ceu.es; 2Facultad de Medicina, Instituto de Medicina Molecular Aplicada (IMMA), Universidad San Pablo-CEU, 28668 Madrid, Spain; 3Centre for Metabolomics and Bioanalysis (CEMBIO), Facultad de Farmacia, Universidad San Pablo-CEU, 28668 Madrid, Spain; antogar@ceu.es; 4Faculty of Experimental Sciences, Universidad Francisco de Vitoria, 28223 Madrid, Spain; noemi.garcia@ufv.es; 5Brain Tumor Laboratory, Fundación Vithas, Grupo Hospitales Vithas, 28043 Madrid, Spain

**Keywords:** glioma, oncolytic virus, glioblastoma, virotherapy

## Abstract

Glioma tumors are one of the most devastating cancer types. Glioblastoma is the most advanced stage with the worst prognosis. Current therapies are still unable to provide an effective cure. Recent advances in oncolytic immunotherapy have generated great expectations in the cancer therapy field. The use of oncolytic viruses (OVs) in cancer treatment is one such immune-related therapeutic alternative. OVs have a double oncolytic action by both directly destroying the cancer cells and stimulating a tumor specific immune response to return the ability of tumors to escape the control of the immune system. OVs are one promising alternative to conventional therapies in glioma tumor treatment. Several clinical trials have proven the feasibility of using some viruses to specifically infect tumors, eluding undesired toxic effects in the patient. Here, we revisited the literature to describe the main OVs proposed up to the present moment as therapeutic alternatives in order to destroy glioma cells in vitro and trigger tumor destruction in vivo. Oncolytic viruses were divided with respect to the genome in DNA and RNA viruses. Here, we highlight the results obtained in various clinical trials, which are exploring the use of these agents as an alternative where other approaches provide limited hope.

## 1. Introduction

Diffuse gliomas are the most frequent central nervous system (CNS) tumors with an infiltrative growth pattern which includes astrocytoma, oligodendrogliomas, and oligoastrocytomas [1]. These malignant tumors are classified by histology and molecular features established by the World Health Organization (WHO) [2]. Glioblastoma (GBM), categorized as WHO grade IV, is the most common and lethal glioma with an incidence of 4.32 per 100,000 habitants in the USA [3]. Since 2005, the treatment guidelines involve a combination of surgical intervention, radiotherapy, and chemotherapy based on the DNA alkylating agent temozolomide (TMZ) [4]. However, new radiotherapy regimens such as tumor treating fields (TTFields) have increased the overall survival by some months. Despite these aggressive therapies, unfortunately, most of the tumors relapse, and the majority of GBM patients die within 21 months [5]. Different factors are responsible for treatment failure, such as a high invasive and infiltrative potential, several resistance pathways, and high intra- and inter- tumoral heterogeneity [6]. The presence of a subpopulation of cancer stem cells (CSCs) appears to be responsible for the tumor cell dissemination through the normal brain parenchyma [7], which contributes to gliomagenesis and recurrence [8].

Even though several strategies are being studied to overcome therapeutic resistance, the second- line treatment has not been well established, and different approaches are being tried [9]. The use of antiangiogenic agents such as bevacizumab are able to add some quality of life but fail to significantly increase patient’s overall survival (OS) [10]. Furthermore, there are no new Food and Drug Administration (FDA) validated therapies for GBM, and all the current alternatives continue in research phases [11].

Many of the ongoing studies are validating the efficacy of immunotherapies including antitumor vaccine-based treatment, immune checkpoints, and viral therapy. Virotherapy is considered a promising strategy for cancer treatment and can be divided into two different approaches: the use of non-replicating viruses as gene delivery vector systems and the oncolytic replicating viruses [12]. The oncolytic virotherapy (OV) that uses replicative viruses amplifies the viral progeny and the danger-associated molecular patterns (DAMPs) which trigger innate and adaptative immune responses [13]. Tumor infection triggers both an antiviral and an antitumor-specific immune response, aiming at stimulating tumor destruction through the induction of specific immunogenic cell death (ICD). OV can also be selected or engineered to be tumor-specific by genetic modifications that limit their pathogenicity and/-or enhance tumor immunogenicity [14].

Upon viral infection, the host cell recognizes specific patterns of the virus known as pathogen-associated molecular patterns (PAMPs) by pattern recognition receptors (PRRs). These receptors initiate the innate immune response, inducing signaling pathways that lead to the expression of interferon (IFN)-β and proinflammatory cytokines such as interleukin (IL)-6, tumor necrosis factor (TNF)-α, or IL-1β, among others, which promote an antiviral state in the tumor environment [15]. However, it is known that some cancer cells are deficient in triggering this immune-mediated response. These kinds of tumor cells are more susceptible to viral replication mediated oncolysis [16]. Viral infection can also promote the antiviral response affecting the tumor microenvironment. The balance between direct tumor destruction by virus replication and the virus-mediated antitumoral immune-response determines OV effectiveness.

There are many hurdles to overcome for the successful clinical application of OVs, such as the immunosuppressive GBM microenvironment or the presence of the blood brain barrier. In fact, the high number of infiltrating immune cells and the cancer stem cells impairs selective viral replication [17] Some patients could have attacked immune memory against specific OVs from previous infections or vaccinations, which could restrict the oncolytic viral therapy. For example, systemic neutralizing antibodies could limit the access of therapeutic viruses to the tumor microenvironment, thus most of them need intratumoral administration [18].

Here, we review the state-of-the-art oncolytic virotherapies for the treatment of brain tumors (Figure 1). We present the main viruses proposed for brain tumor oncolytic therapy alone and in combination with other therapeutic approaches. We focus on viruses used in preclinical studies (Table 1 and Table 2) and clinical trials (Table 3) performed mostly in GBM patients.

## 2. DNA Viruses Proposed as Glioma Oncolytic Agents

### 2.1. Herpes Simplex Virus Type I

Herpes simplex virus Type 1 (HSV-1) is an enveloped double stranded DNA virus that belongs to the *Herpesviridae* family. This virus is known for its ability to infect and replicate in neural tissue, making it a candidate for glioma treatment. Natural HSV-1 entry is mediated by the binding of viral glycoprotein D (gD) to the cell surface protein CD111, also known as Nectin-1 [19]. Nectin-1 is differentially expressed in gliomas as compared to normal tissue [20]. It has been demonstrated that this virus mediates a direct lytic effect in tumor cells, and, in addition, most of the in vivo effects suggest a tumor destruction mediated by activation of tumor-specific immune responses [21].

As a neurotropic virus, HSV-1 presents through some toxicity and potentially may present side effects associated with normal tissue infection. The genetic attenuation of the virus can overcome this problem, allowing the introduction of some therapeutic genes.

#### 2.1.1. Herpes Simplex Virus-1 Pre-Clinical Research

A first approximation for oncolytic attenuation of HSV-1 was *dls*ptk HSV. This virus lacks thymidine kinase protein (TK), which is a key protein needed for viral replication in non-dividing cells. *dls*ptk HSV showed effectivity by killing glioma cells in vitro and prolonging survival in both subcutaneous and orthotopic in vivo models. A serial concern regarding this modified virus for tumor treatment is the possibility of a progressive infection in immunocompromised patients [22]. To avoid this possibility, other strategies have been developed. HSV-1716 is a modified HSV-1 strain 17 with a deletion of 759 bases in γ_1_34.5 loci. γ_1_34.5 is a viral antagonistic protein known to block protein kinase R (PKR) antiviral signaling in infected cells. This deletion was shown to prevent encephalitis in mice infected with the mutant virus by eliciting an abortive infection in non-tumoral cells [23]. This virus has demonstrated infectivity and oncolytic activity in cell lines and patient-derived glioma cells [24].

A step forward in HSV-1 attenuation for oncolytic use was HSV-1 G207. This virus is a double mutant constructed by the insertion of *Escherichia coli* lacZ gene into the coding sequence for viral ICP6 gene and deletion of both copies of γ_1_34.5 loci within the viral genome [25]. ICP6 is a ribonucleotide reductase essential for viral replication and growth in non-dividing cells. By eliminating this gene, HSV-1 G207 becomes tumor restricted. This virus showed no neurovirulence after intracerebral injection of 1 × 10^7^ pfu both in mouse and in owl monkeys. This OV also demonstrates extended survival and lower tumor growth ratio in mice [26].

Although deletion of γ_1_34.5 loci and ICP6 strongly improves viral tumor restriction, it also attenuates virus replication and treatment efficacy [27]. In order to reduce this effect, a modified version of HSV-1 named rQNestin34.5 was designed to leave only the carboxyl terminus of ICP6 linked to GFP, deleting both copies of γ_1_34.5 loci and reinserting a copy of γ_1_34.5 under the glioma specific upregulated gene nestin promoter. This virus presented an increased oncolytic effect, higher viral replication, and more efficient propagation in comparison with a control virus. In vivo models treated with rQNestin34.5 resulted in an increase of mice survival and tumor reduction as compared with control HSV virus [28]. In order to move along into a clinical phase, the carboxyl terminus of ICP6 was completely deleted due to its partial neurotoxicity originating from the rQNestin34.5v2 virus. This virus displayed no neurotoxicity in immunocompetent mice [29].

NG34 was developed in a very similar way to rQNestin34.5. This virus also has deletion of ICP6 and both γ134.5 loci, but, in this construction, the human GADD34 gene was placed under the Nestin promoter. The carboxyl terminus of GADD34 shares homology with the viral protein ICP34.5 carboxyl-terminus, being the domain responsible for eIF2α dephosphorylation, but is lacking from the neurotoxicity of ICP34.5. This change reduces the toxicity of this virus in normal cells where there is a low amount of Nestin promoter activation. NG34 showed similar virus titers and cytotoxicity in vitro, similar survival in vivo as compared to rQNestin34.5, and less neurotoxicity in immunocompetent mice [30].

A step forward is the development of NG34scFvPD-1. This virus is a modified version of NG34 that also expresses a single chain antibody against mouse PD-1 under the control of a cytomegalovirus (CMV) immediate early (IE) promoter. PD-1 has a key role in inhibiting adaptive immune response against tumor cells. Anti PD-1-based therapies have been shown to promote antitumor responses in multiple studies. NG34scFvPD-1 has similar oncolytic properties as NG34 in vitro and demonstrated an improvement in animal survival in immunocompetent mice as compared to NG34, generating antitumor immune-memory that protected these mice against a reimplantation of tumor cells [31].

Following a different strategy, HSV-1 C134 was developed as a chimeric virus that includes not only the deletion of γ_1_34.5 loci but also the expression of human cytomegalovirus (HCMV) IRS1 gene. This insertion increases viral replication and lytic effect when administered intrathecally in a murine GBM tumor model inducing antitumor T cell mediated immune responses, which elicit long systemic immune-memory enhancing survival [32,33].

HSV-1 G47Δ is a triple mutant virus based on G207 to increase the oncolytic effect. G47Δ includes a deletion in α47 gene and herpes unique short 11 (US11) promoter. α47 is a viral antagonistic factor that inhibits MHC-I presentation. This deletion increases viral infection immunogenicity. As in wild type (WT) HSV, α47 is followed by US11 and deletion of α47 gene, and US11 promoter places the lytic factor US11 under control of immediate-early α47 promoter, increasing tumor lysis after virus infection. HSV-1 G47Δ has enhanced viral growth and displays a higher tumor lytic effect after intracerebral administration inoculation in mice [34].

A different strategy to improve tumor-specific oncolytic activity is retargeting HSV to a tumor specific protein by modifying the gD entry protein. HSV-1 R-LM113 is a recombinant virus with an insertion in gD of a single chain of an antibody against HER2. HER2 is a marker of bad prognosis that is present in up to 80% of human gliomas. This virus showed no neurotoxicity in HSV sensitive mice and doubled survival in both tumor established and HER2 overexpressing in vivo models [35].

Decrease of tumor regrowth after oncolytic treatment is the objective followed by rapid antiangiogenesis mediated by oncolytic virus (RAMBO). This virus includes deletion of γ_1_34.5 loci, GFP linked to ICP6 carboxyl terminus, and expression of human Vstat120 gene under immediate early IE4/5 HSV promoter. Vstat120 encodes for the extracellular fragment of brain-specific angiogenesis inhibitor 1 (BAI1) and has a potent antiangiogenic and antitumor effect. RAMBO has shown an increase in survival and tumor reduction in vivo compared with a control HSV and a reduction in the vascular volume fraction in the tumor due to delivery of Vstat120 [36].

Cytokine expressing HSV-1 is another approximation that has been explored in different studies for glioma treatment. HSV-1 expressing murine IL-12 in substitution of γ134.5 (M002) has been compared with other strains of HSV-1 such as R3659 and G207 in the intracranial 4C8 glioma mouse model, demonstrating an increase in animal survival, a higher tumor infiltration of CD4+, CD8+, and natural killer (NK) cells, and a longer persistence of virus titers inside the tumor [37]. Similar results were obtained with HSV-1 R8306 in which γ134.5 genes were replaced by murine IL-4. This virus induced a higher infiltration of macrophages CD4+ and CD8+ in the tumor and longer survival in mice in comparison to HSV-1 R8308, a virus expressing the anti-inflammatory cytokine IL-10, or a control virus R3616 [21] (Table 1). The pre-clinical use of these genetically modified viruses has demonstrated an increase in HSV tumoral selectivity and an enhancement in some immune evasion gene expression. For these reasons, modified HSVs have been assessed for glioma treatment in clinical studies.

#### 2.1.2. Herpes Simplex Virus-1 Clinical Studies

Phase I and Ib clinical studies have demonstrated patient dose tolerance to HSV-1716 of up to 10^5^ virus pfu, and in two different studies in which four out of nine and three out of twelve patients survived for more than 1 year [38,39]. A phase II study with two patients has been completed, but results are have not yet been published (NCT02031965).

Phase I and Ib clinical studies in HSV-1 G207 demonstrated no neurovirulence in GBM patients, even at high intracerebral inoculation doses (3 × 10^9^ pfu) [40,41]. Moreover, its administration with radiotherapy in nine recurrent GBM patients showed a good response in six of them [42].

rQNestin34.5v2 virus is now under a phase I clinical trial with 108 glioma patients. This virus is administered in combination with cyclophosphamide, an immunomodulating drug that has been shown to promote a virus replication increase in tumors and improved patient survival (NCT03152318) [43].

HSV-1 G47Δ phase I clinical trial showed limited toxicity, and a phase II trial resulted in increased survival of treated patients [44].

Another phase I trial in 24 recurrent GBM patients is being conducted with HSV-1 C13, but no results have been presented (NCT03657576) (Table 3).

### 2.2. Adenovirus

Adenoviruses are icosahedral non-enveloped viruses with a double-stranded DNA genome. In total, 57 serotypes have been described in humans, some causing pathologies. In addition, other adenovirus serotypes infect different mammal species. Adenoviruses have been studied for decades, being an interesting viral vector for gene delivery. Cell tropism of human adenoviruses (HAd) differs between different serotypes. C subgroup of HAd, formed by serotypes 1, 2, 5, and 6, causes mostly respiratory infections, and it is known that virus cell entry is mediated by chimeric antigen receptor (CAR), heparan sulfate proteoglycan (HSPG), major histocompatibility receptor 1 (MHC-I), vascular cell adhesion molecule 1 (VCAM-I) and Integrins as receptors. Adenovirus serotype 5 (Ad5) is the most studied and used as a gene delivery tool [45]. Characterization of the virus genetic elements and the possibility of manipulating them have allowed the generation of recombinant viruses, enabling the development of several oncotherapeutic options.

Until now, the best approximations into the use of replicative adenovirus as oncolytic therapy have been the conditionally replicative adenoviruses (CRad). Different generations of CRad have been developed in recent years with promising results in gliomas in both pre-clinical and clinical studies [46].

#### 2.2.1. Adenovirus Pre-Clinical Research

The first generation of CRad started with Onyx-015, a chimeric adenovirus generated from two and five serotypes that has a deletion in the *E1B-55kD* gene and was approved in China for the treatment of head and neck cancer in 2005 [47]. E1B-55kD protein binds and inhibits p53 in infected cells allowing viral replication. Due to this modification, Onyx-015 is deficient for replicating in non-tumor cells [48]. This virus showed a powerful antitumoral effect in both p53 wild type (wt) and p53 mutant glioma xenograft mouse tumor models, inducing a relevant tumor regression [49].

A second generation of CRad improved the initial attempts in order to not only decrease the infectivity of adenovirus in non-tumor cells but also to increase tumor infectivity. Delta-24 is an Ad5 with a mechanism different to Onyx-015 to restrict replication in tumor cells. This virus has a 24-base-pair deletion in E1A gene, a protein that binds to the tumor suppression protein Rb. Delta-24 lacks this ability, which contributes to the restriction of virus replication to Rb.deficient tumor cells. This virus is also modified to incorporate an Arg-Gly-Asp (RGD) tripeptide. This peptide recognizes integrins present in gliomas, facilitating viral entry into the tumor cells [50,51]. This modified virus is known as Delta-24-RGD or DNX-2401 [52], and its administration demonstrated a higher antitumoral effect in tumors with a lack of Rb pathway, while wt cells remained resistant to infection in in vivo studies [53].

Although partial deletion of early gene E1A makes the virus more selective for Rb lacking cells, excessive accumulation of E1A protein can induce toxicity in normal cells. To overcome this situation, a recombinant virus was made, inserting the cell cycle dependent E2F-1 promoter as the regulatory promoter for E1A gene. Thus, fast replicating tumor cells preferentially express E1A under E2F-1 promoter as compared to no-replicant normal cells. This adenovirus, which also includes 24-base-pair deletion of E1A and a RGD motive to improve tumor infectivity, is known as ICOVIR-5 [54]. In addition, ICOVIR-5 showed less percentage of normal cells infected and stronger antitumoral effect as compared to Delta-24 and Delta-24-RGD in vitro. An orthotopic murine model of U87 tumor cell xenografts treated with ICOVIR-5 demonstrated longer survival than no treatment as well as a comparable survival rate to Delta-24-RGD [53]. Further modifications of this virus, such as ICOVIR-7 or ICOVIR-15, have improved tumor-specific cytotoxicity. ICOVIR-7 is a modified ICOVIR-5 that includes four palindromic E2F-1 sites in the promoter instead of one, increasing E1A expression in tumor cells [55]. ICOVIR-15 includes an Sp-1-binding site in the promoter to redirect E1A-Δ24 transcription towards pRb deregulation, increasing the tumor viral replication [56]. More relevant in the treatment of gliomas is ICOVIR-17 adenovirus, which, in addition to the already mentioned modifications, expresses a soluble form of the human sperm hyaluronidase (PH20) regulated under the major viral late promoter (MLP) of adenovirus in order to decrease the amount of hyaluronic acid in the tumor environment [57]. Hyaluronic acid is an abundant element of the tumor matrix. Hyaluronic acid in tumors is associated with metastasis in the brain and inhibition of infiltration antitumor treatments [58,59]. In comparison with ICOVIR-15, ICOVIR-17 showed potent oncolytic activity in vitro as well as an increased survival in a murine GBM tumor model [60]. VCN-01 is a modified ICOVIR-17 in which RGD motive has been relocated into the fiber shaft protein of the virus in order to increase infectivity. This virus shows a potent oncolytic activity against aggressive infiltrative and non-infiltrative tumors both in vitro and in vivo [61].

Immune stimulation is another strategy that has been explored using adenoviruses to induce oncolysis and tumor regression. Delta-24-RGDOX is a modified Delta-24-RGD adenovirus that expresses the immune stimulatory OX40 ligand (OX40L) to stimulate antigen presentation in tumor cells by recruiting and activating tumor-specific T cells [62].

Delta-24-GREAT (glucocorticoid receptor enhanced activity of T cells) follows a very similar aim to Delta-24-RGDOX. This virus is a modified version of Delta-24-RGD virus that expresses murine glucocorticoid-induced of tumor necrosis factor receptor (TNFR) family-related gene ligand (GITRL). This ligand is mainly expressed by antigen presenting cells (APCs) and has a co-stimulatory effect on CD4 and CD8 lymphocytes inducing activation [63]. Delta-24-RGDOX showed an efficient CD4+ and CD8+ activation in pre-clinical models [62]. Delta-24-GREAT elicited antiglioma specific immune response in an immunocompetent model, increasing mice survival and developing immune memory that protected animals from a tumor rechallenge [63]. Lastly, treatment of immunocompetent mice with intratumoral administration of Ad-RTS-IL-12 combined with oral administration of veledimix resulted in an increase of CD8 T lymphocyte infiltration, a decrease in tumor growth, extended animal survival, and antitumor immune specific memory [64].

With the same objective, activating CD4 and CD8 antitumor specific response, adenoviral vector Ad-RTS-IL-12 was created. In this vector, expression of mIL-12 is regulated under the RheoSwitch Therapeutic System^®^ (RTS^®,^ Ziopharm oncology IN, Boston, MA) to deliver the cytokine only in the presence of a specific ligand. This system is a modification of ecdysone receptor (EcR), an inducible gene regulation system originally present in insects. The activator that triggers IL-12 production in this vector is a synthetic analog of the insect molting hormone ecdysone called veledimex. Veledimex can pass through the blood brain barrier and reach brain tumors, making oral administration possible [64] (Table 1).

The use of CRad as a therapeutic option in GBM treatment has additional possibilities to be explored. Recent advantages in the use of high capacity adenoviral vectors provides an interesting platform to deliver therapeutic genes to the tumor environment [65]. Either alone or in combination with other treatments, the use of these non-replicating viral vectors is an interesting tool that needs additional attention. At present, second and third generations of oncolytic adenovirus have achieved effectiveness in pre-clinical models with no serious adverse effects; these results are sufficiently encouraging to make adenovirus a real approach for glioma treatment.

#### 2.2.2. Adenovirus Clinical Studies

A phase I clinical trial using Onyx-015 was carried out with 24 glioma patients showing no adverse effects and regression in one patient and no progression of the disease in another participant [66]. Several phase I and II clinical trials have been carried out using both the Delta-24 virus and the Delta-24 RGD version with promising results. Delta-24-RGDOX is now in a phase I clinical trial with 24 GBM patients (NCT03714334). Intratumoral administration of Ad-RTS-hIL-12 is now under a phase I/II clinical trial in patients with pediatric brain tumors in combination with oral administration of veledimix (NCT03330197) (Table 3).

### 2.3. Vaccinia Virus (VV)

Vaccinia is an enveloped double-stranded DNA virus belonging to the *Poxviridae* family. VV is a classic virus that made possible one of the greatest medical milestones, the eradication of smallpox. Viral entry is not dependent on cell receptors but on membrane fusion, allowing this virus to infect almost any mammalian cell type. This natural tropism for almost any tissue, along with its fast and efficient non-integrative replication cycle, its cell-to-cell spread ability, and its relatively easy genetic modification, makes VV an interesting tool in generating a recombinant virus as well as in designing alternative approaches for glioma oncolytic treatment [67,68].

#### 2.3.1. Vaccinia Virus Pre-Clinical Research

Strategies to increase the virus oncolytic potential focus on enhancing apoptosis, as the recombinant rVV-p53 has shown a greater ability to trigger apoptosis in both in vitro glioma cells and animal tumor models compared with wt VV [69]. IL-12 and IL-2 are two important cytokines involved in the activation of a robust Th1 immune response. Recombinant viruses expressing these cytokines in general present an effect in halting tumor growth and promoting the antitumor specific activation of the adaptive immune response. In order to avoid toxicity, the recombinant VV expressing these cytokines must be administered in a very low dose (10^2^–10^3^ pfu) [70].

Combination of different recombinant VV is another approach that has been explored in GBM treatment. Coinfection with high doses of rVV-p53 (2 × 10^7^ pfu) with a low dose of a recombinant rVV-mIL12 (10 pfu) resulted in a strong tumor inhibition with an increase in the immune response after intratumoral injection in a nude mouse glioma model [71]. However, this strategy must be validated in an immune competent animal tumor model to determine its full potential. A double-deleted version of western reverse (WR) VV strain, also known as vvDD, has been developed in order to increase cell lysis and at the same time limit the growth of this aggressive strain into tumor cells [72,73]. vvDD lacks thymidine kinase protein (TK), which determines virus dependence on dividing cells to replicate, and vaccinia growth factor (VGF), a secreted protein that primes surrounding cells for division and VV infection [72]. This virus has demonstrated an efficient destruction of rat and human malignant glioma tumor cells in vitro. Systemic delivery was able to reach solitary and multifocal tumors, increasing surveillance in animals [74]. A safety dose assay in non-human primates has proven that vvDD has no adverse effects in contrast with the WR unmodified strain that produced several complications, such as fever, skin rash, or the presence of virus in multiple organs [73]. Combination of vvDD with other GBM treatments, such as rapamycin or cyclophosphamide, seems to increase the oncolytic potential of the virus [74]. A modification of this virus expressing IL15Rα (vvDD-IL15Rα), aimed at boosting the immunostimulating effect of the virus in combination with the direct lytic effect, has been proven to be quite efficient in killing murine glioma cells in vitro. Intratumoral administration of this virus results in prolonged survival and a significant recruitment of NK and CD8+ T cells into the tumor. Secondary effects such as ventriculitis-meningitis were observed in some animals after the treatment [75]. A different strategy in the use of VV as an oncolytic virus is the combination of direct effect and a specific drug delivery system. Following this approach, TG6002 was developed as a double-deleted recombinant VV virus which has been tested for the treatment of gliomas. This virus lacks TK and ribonucleotide reductase genes, which allows the virus to replicate mainly in tumor cells. In addition, TG6002 has been modified to express the yeast FCU1 gene. This gene encodes cytosine deaminase and uracil phosphoribosyl transferase, which transforms the pro-drug flucytosine (5-FC) into cytotoxic 5-fluorouracil (5-FU) and 5-fluoro-uridilyl monophosphate (5-FUMP).

TG6002 virus can replicate in glioma cells and induce cell death in vitro. In addition, its systemic administration in an orthotopic brain tumor mouse model showed an increase in animal survival. These results were improved with oral administration of 5-FC pro-drug [76] (Table 1).

#### 2.3.2. VV Clinical Studies

Despite the high number of preclinical studies in VV confirming the safety and the oncolytic activity, it has not yet transferred into successful clinical trials, and only TG6002 is being evaluated. The main study consists of a phase I clinical trial with 78 GBM patients using a combination of TG6002 and 5-FC. A total of 24 patients of this preliminary study without tumor progression will be included in the next phase IIa study (NCT03294486) (Table 3).

### 2.4. Myxoma

Myxoma virus (MYXV) is an enveloped double-stranded DNA that belongs to the poxvirus family. This virus is highly pathogenic in European rabbits and has not been described as producing disease in other vertebrates [77]. However, this virus can infect and replicate in cells displaying deficiencies in the IFN system, making it a good candidate for oncolytic treatments [78].

MYXV has shown ability to infect and destroy cells in many human and rat glioma cell lines, some of them being partially resistant. This virus demonstrated safety characteristics and showed tumor regression in an in vivo mouse model [79]. Part of the effect of MYXV is produced by a decrease in MHC I expression in infected glioma cells and the consequent NK elimination [80]. One disadvantage of using MYXV as oncolytic virus in vivo is its poor ability to proliferate outside the tumor injection area [79]. Oncolytic activity in this context relies on a tumor specific immune stimulation upon virus injection.

#### Myxoma Virus Pre-Clinical Research

Different approaches have been used to increase the expansion and the effectivity of this virus. One is radiotherapy and TMZ pre-treatment followed by infection with MYXV, resulting in increased spread of MYXV infection and decreased cell viability in GBM cell lines as compared to other non-tumor cells [81]. The use of MYXV replication permissive cells as a delivery tool for the virus in the tumor is another strategy used to colonize gliomas with the virus. Adipose-derived stem cells (ADSCs) are susceptible to MYXV replication without compromising cell viability. These cells are an excellent vehicle for acting as a constant supplier of MYXV in tumors. Co-cultured myxoma virus infected ADSCs with GBM cell lines results in a widespread infection and low viability of tumor cells. Intrathecal treatment of orthotopic GBM mouse models with MYXV-ADSCs but outside of the tumor site resulted in tumor infection and increased animal survival [82].

MYXV-M011L-KO is a modified MYXV lacking the antiapoptotic protein M11L. This virus showed a potent apoptotic effect in patient-derived GBM CSCs as compared to the wt MYXV. MYXV-M011L-KO intratumor administration in a GBM tumor model of immunocompetent mice showed a synergistic effect with temozolomide co-treatment in prolonging animal survival [83] (Table 1).

These findings together with further information from ongoing studies have a potential to make MYXV a viable option for brain tumor clinical management.

### 2.5. Parvovirus

*Parvoviridae* is a family of viruses with a single-stranded DNA genome. These viruses have an icosahedral capsid. Thus far, 134 different parvoviruses have been described as being able to infect several animal species [84]. The rat protoparvovirus H-1 known as H-1PV is non-pathogenic in humans and binds to host cell surface receptors entering into cells by clathrin-mediated endocytosis [85]. H-1PV DNA replication occurs when active mitotic cells enter into S-phase [86], triggering a DNA damage response and cell-cycle arrest that finally kill target cells.

#### 2.5.1. Parvoviridae Pre-Clinical Research

It has been proposed that H-1PV destroy cisplatin and TNF-related apoptosis-inducing ligand (TRAIL) resistant glioma cells by inducing cathepsins B and L aggregation and decreasing the expression of their inhibitors, the cystatins B and C [87]. Complete GBM tumor regression was observed in rat models using H-1PV in an early tumor infection by Geletneky and colleagues [88,89]. In addition, a selective replication and a lack of toxicity in the oncolytic LuIII parvovirus and the minute virus of mice (MVV) have been observed [90,91,92,93] (Table 1).

#### 2.5.2. Parvoviridae Clinical Studies

Following this promising data, a phase I/IIa clinical trial was conducted in 18 unifocal recurrent GBM patients. Participants were divided into two groups. First, one of the groups was treated intravenously, and the other group received an intratumoral injection. The second H-1PV administration (into the tumor cavity during surgery) was the same in both groups (ParvOryx01: NCT01301430) [94]. As an initial approach, the safety and the tolerance were evaluated, indicating a lack of toxicity, and, upon intravenous administration, the virus was able to cross the blood brain barrier and reach the tumor [95]. Additionally, H-1PV was able to enhance immunogenicity within the tumor microenvironment [96]. H-1PV treated patients displayed an increase in tumor-infiltrating cytotoxic T cells, induction of cathepsin B, and the expression of IFN-γ and IL-2, among other cytokines within the tumor microenvironment (Table 3).

The notable safety and tolerability in systemic and local administration makes *Parvoviridae* a good viral family candidate to treat GBM combined with other immunotherapeutic agents.

## 3. RNA Viruses Proposed as Glioma Oncolytic Agents

### 3.1. Reovirus

*Reoviridae* is a family of double-stranded RNA non-enveloped viruses that can cause asymptomatic or mild enteric infections in humans. Orthoreovirus, also known as reovirus, has been shown to be a natural oncolytic virus because it can overtake and specifically replicate in Ras pathway activated cells, which are commonly present in gliomas [97,98].

#### 3.1.1. Reovirus Pre-Clinical Research

Reovirus treatment of subcutaneous and intracerebral glioma mouse models resulted in an intense and often total regression [99]. Reovirus-mediated oncolysis has been tested in preclinical models inducing a direct tumor lysis, an increase of T cell infiltration, together with a higher expression and secretion of Type I IFN in the tumor microenvironment [100] (Table 2).

#### 3.1.2. Reovirus Clinical Studies

Single reovirus intratumoral administration in 12 recurrent glioma showed no adverse effects in a phase I clinical trial [101]. Similar observations were obtained using an intratumoral infusion of 72 h in 18 participants [102]. Recently, a phase Ib clinical trial posted an increase in tumor leukocyte infiltration and higher expression of IFN, caspase 3, and programmed death-ligand 1 (PD-L1) in tumors from reovirus treated patients [100] (Table 3).

Despite these preliminary safety and tolerability results, there are currently no available clinical trials assessing the impact in the patient’s survival.

### 3.2. Measles

Measles virus (MV) belongs to *Paramixoviridae*, a family of enveloped viruses with a negative single-stranded RNA genome. MV fusion (F) and hemagglutinin (H) proteins have been demonstrated to play a role in the antitumor activity of the virus in gliomas [103]. In this sense, MV Edmonston´s vaccine (MV-Edm) is one of the approximations that have been considered for glioma treatment. MV enters cells by interaction of the viral H protein with the cell receptor CD46, a protein present in almost all human cells and overexpressed in tumor cells [104].

#### 3.2.1. Measles Pre-Clinical Research

MV-Edm was modified to express carcinoembryonic antigen (MV-CEA) in order to track viral gene expression in vivo through blood analysis, since this factor can be released and detected in blood [105]. Glioma cell infection with MV-CEA leads to a syncytial formation mediated apoptosis, while normal cells do not develop a cytopathic effect. Animal models showed a significant increase in surveillance and tumor regression after intratumoral treatment with MV-CEA [105].

MV-NIS is another modification of the MV-Edm, in this case the recombinant virus expresses the human sodium iodide symporter (NIS) to improve the monitoring of MV infection in vivo in brain tumors with a non-invasive method by using systemic administration of ^123^I, ^124^I, ^125^I, or ^99m^Tc isotopes and measuring the isotope accumulation in virus-replicating cells. *MV-NIS* increases cytopathic effect of MV treatment through radiotherapy by local accumulation of ^131^I. MV-NIS induced longer survival in mouse models and increased viral titers and cell death in comparison with MV-CEA [106].

In addition, MV-GFP-H_AA_-scEGFR is a recombinant virus modified to ablate H protein recognition by the two natural viral receptors CD46 and signaling lymphocytic activation molecule (SLAM). Instead, this virus expresses a single chain antibody that binds to epidermal growth factor receptor (EGFR) fused to the C terminal end of the virus H protein. Amplification of EGFR is one of the most frequent genetic alterations in GBM. This characteristic determines the specificity of MV-GFP-H_AA_-scEGFR. In vitro and in vivo experiments with MV-GFP-H_AA_-scEGFR displayed results similar to MV-GFP, with significant regression and induction of cell apoptosis. However, administration of MV-GFP-H_AA_-scEGFR at the central nervous system in CD46-expressing mice resulted in no neurotoxicity [107].

MV-141.7 and MV-AC133 are two other recombinant viruses in which the H protein has been modified to retarget the virus to the CD133 receptor. CD133 is a marker commonly expressed by GBM CSC. MV-141.7 resulted in a better survival rate in comparison with MV-Edm in the treatment of a orthotopic glioma mouse model [108] (Table 2).

#### 3.2.2. Measles Clinical Studies

A phase I clinical trial using MV-CEA was carried out in 23 recurrent participants. One group was treated directly in the resectioned cavity, and the other patients were also treated before the surgery by catheter. They observed a tolerance up to 10^7^ pfu Tissue culture infectious dose 50 (TCID50) and no significant differences between groups in progression free survival (PFS) at 6 months (NCT00390299) (Table 3).

One of the main limitations in using MV in brain tumor patients is the invasive intratumoral administration procedure due to the low efficacy when the virus is administered intravenously [109].

### 3.3. Vesicular Stomatitis

Vesicular stomatitis virus (VSV) is an enveloped negative strain RNA virus that belongs to the *Rhabdoviridae* family. VSV entry is mediated by its glycoprotein spike (G) and a very ubiquitous cell receptor, the low-density lipoprotein receptor (LDL-R), which allows the virus to enter almost every cell type [110]. Moreover, this virus has a short replication cycle of around 3 h, leading to a cytopathic effect that can be observed as early as 4-6 h after infection, making it a good candidate for treatment in a wide range of tumors [111].

#### Vesicular Stomatitis Virus Pre-Clinical Research

Unmodified VSV is able to kill a large variety of tumor and immortalized cells in vitro as well as inhibit the growth of C6 GBM cells in flanks of mice [112]. However, the virus can be lethal for animals if, upon infection, they do not mount an efficient IFN response. In this context, the virus toxicity can be contained by administration of recombinant type I IFNs without blocking the oncolytic effects on tumor cells [113]. In order to reduce unspecific neurotoxicity, several viral modifications have been developed, such as the deletion of the G encoding gene from the viral genome in the VSV-ΔG viral vector. The cytopathic effect of this vector in glioma cells is markedly lower than unmodified VSV and does not reach as many cells as other recombinant viruses [114].

VSV^ΔM51^ is an attenuated replicating virus strain. This strain has a single amino acid deletion in the matrix protein (M), affecting the nuclear-cytoplasmic transport. This modification impedes the M protein ability to block the IFN-β mRNA transport from the nucleus to the cytoplasm and thus affects the antagonistic activity of IFN, limiting the virus replication to tumor cells incapable of producing IFN, allowing infected normal cells to produce a normal IFN response and therefore limiting the virus spread. This virus showed oncolytic activity against 14 glioma cell lines and 15 primary human tumor glioma cells, infecting and inducing cell death in all of them. In vivo experiments showed tumor regression and prolonged survival in U87 and U118 mouse tumor models [115].

A different strategy to attenuate the virus is the tumor-adapted VSV-rp30 (repeated passage 30). In this case, this VSV was adapted to glioma cells by 30 serial passages in which the time between infection and virus recovery was reduced after every 10 passages in order to select the fastest replicant viruses. VSV-rp30 has a higher replication rate on glioma cell lines and less cytotoxicity in non-tumor cells as compared to wild type virus [92]. The virus can infect in vivo models after intravenous administration and destroy GBM brain tumors with tumor dissemination [116].

Another strategy to attenuate the virus is the modification of the genome in order to reduce the cytoplasmic tail of the G protein [117]. VSV-CT9 and VSV-CT1 are truncated G protein versions of the virus that have reduced the cytoplasmic region of the G protein from 29 amino acids to nine and one, respectively [118]. These two versions showed efficacy to kill in vitro GBM cells, VSV-CT9 being more toxic for normal cells as compared to VSV-CT1. Viral toxicity for normal cells can be reduced by a co-treatment with IFNα, inhibiting viral replication in these cells, while viral titers remain high (with a small decrease) in glioma cells [114]. VSV-CT1 also showed less neurotoxicity after intracranial injection and intranasal inoculation as compared to wt VSV [119].

A combination of two different strategies of virus attenuation resulted in VSV-CT9-M51. This mutant combined both strategies used to develop VSV ^ΔM51^ and VSV-CT9. VSV-CT9-M51 showed less neurotoxicity in normal cells than both VSV-CT9 and VSV ^ΔM51^ while retaining the ability to infect, spread within, and kill human GBM in a mouse model after systemic administration, also triggering higher type I IFN dependent responses in the animals [119].

Another attenuation strategy that has been proposed to reduce virulence of VSV is gene rearrangement [120,121]. Introducing foreign genes such as GFP or RFP at the first position of the viral genome is a different way to attenuate VSV. VSV-p1-GFP and VSV-p1-RFP showed a high cytopathic effect and induced death after infecting U87 GBM cells in vitro, having at the same time lower toxicity in non-tumor cells. VSV-p1-GFP showed promising results in animal models [114].

Other strategies to reduce adverse effects beyond direct attenuation of the virus have been developed. Co-infection of VSV-CT9-M51 (intracranial) with an adeno-associated virus expressing mouse IFN-β (AAV-mIFN-β) or co-treatment with ribavirin resulted in less neurotoxicity and an overall survival extension in a GBM mouse model [122] (Table 2).

Most of the VSV experiments were done in immunocompromised mouse models, thus future assays assessing the impact in the systemic immune response should be developed to translate these clinical research outcomes into clinical practice.

### 3.4. Newcastle Disease Virus (NDV)

NDV is an enveloped, negative sense, single-stranded RNA virus. This virus belongs to the family *Paramyxoviridae*. NDV mainly infects avian species while having marginal pathogenicity in humans [123]. Depending on the virus pathogenicity in chickens, the different NDV strains can be classified as velogenic, mesogenic, and lentogenic [124]. Following NDV infection, human cells induce the type I IFN response [125].

Although NDV specific tropism for cancer cells is poorly understood, it has been postulated that the small GTPase Rac1, which is involved in the maintenance of GBM stem properties [126], is required for NDV replication [127]. Preferential replication could also be explained by tumor-limited replication due to deficiencies in the type I IFN system present in many GBM patients [128].

#### 3.4.1. Newcastle Disease Virus Pre-Clinical Research

Our group have recently observed that type I IFN-deficient GBM CSCs are more receptive for NDV replication than type I IFN competent cells. This fact was also noted in the mouse model, in which NDV treatment reduced tumor volume only in IFN-deficient bearing cells [129]. Preclinical animal models have reported an apoptotic effect using NDV in GBM treatment [130,131], an increase in the median survival from 28 to 64 months in the mouse models [132], as well as a synergistic effect with TMZ [133] (Table 2).

#### 3.4.2. Newcastle Disease Virus Clinical Studies

Although some studies have been done in small cohorts of GBM patients, all of them have shown promising results. Csatary and colleagues treated intravenously four GBM patients with the mesogenic strain MTH-68/H. They reported an increase in survival rates to 5–9 years, together with an enhancement in the quality of life, after conventional treatment [134]. Similar effects were observed in 10 patients treated with the vaccine Viral Oncolysate-Pulsed DC (VOL-DC) composed of NDV infected dendritic cells [135]. One additional report describes that repeated intravenous administration of the lentogenic NDV strain OV001/HUJ in 14 GBM patients achieved complete tumor regression in one patient [136] (Table 3).

### 3.5. Seneca Valley

Seneca Valley virus isolate 001 (SVV-001) is a non-enveloped positive single chain RNA virus belonging to the *Picornaviridae* family. This virus was isolated and identified from a contamination of culture cells and does not produce any described disease in animals. SVV-001 virus is homologous and is serologically related to 12 swine picornaviruses, thus it is believed to come from a porcine trypsin contamination. SVV-001 has exhibited tropism and oncolytic activity for neuroendocrine tumors [137,138].

#### Seneca Valley Virus Pre-Clinical Research

In vitro experiments with six different GBM CSCs resulted in total infection and significant decrease in viability for four of them. Permissibility of tumor glioma cells is dependent on the presence of α2,3-linked and α2,6-linked sialic acids. Intravenous injection with SVV-001 showed infection, cell lysis, and prolonged animal survival on permissive GBM intracranial xenograft in Rag2 SCID mice models [139].

### 3.6. Poliovirus

Poliovirus belongs to the *Picornaviridae* virus family. These encapsidated viruses have a positive single strand RNA [140]. Poliovirus can cause neurotoxicity, although Gromeier and colleagues eliminated this by replacing the internal ribosome entry site (IRES) of the poliovirus vaccine Sabin strain with the non-virulent human rhinovirus type 2 (HRV2) [141].

#### 3.6.1. Poliovirus Pre-Clinical Research

The resulting PVS-RIPO recombinant virus can infect and reduce glioma cell viability in vitro [142]. In addition, this virus can trigger cytolysis of GBM primary cultures [143]. Finally, PVS-RIPO can halt tumor growth in a murine GBM flank tumor model [144] and increase the mice’s OS after intracranial virus administration [145].

The efficacy of PVS-RIPO appears to be correlated with CD155 expression, which is known to be overexpressed in some cancers, including human GBM, specifically in CD133+ cells [146]. All of this evidence proves that poliovirus is capable of inhibiting GBM tumoral growth in preclinical models (Table 2).

#### 3.6.2. Poliovirus Clinical Studies

An interventional clinical study (NCT01491893) with 61 recurrent GBM patients has shown safety in the intratumoral injection of the virus as well as an increase in patient survival rate of 21% at 36 months as compared to 4% in the historical control group [147]. With the aim of confirming the safety and the test efficacy of the virus, there are currently two active clinical trials for recurrent glioma. An interventional phase II study with 122 enrolled adult patients (NCT02986178) and a phase Ib with 12 malignant glioma children (NCT03043391) are ongoing (Table 3).

### 3.7. Sindbis

Sindbis virus is a small alphavirus of positive stranded RNA genome surrounded by a capsid protein that belongs to the *Togaviridae* family. The natural hosts are birds, but mosquitoes act as vectors to infect mammals, including humans, through their bites [148]. The infection occurs when the virus binds to the 67 kDa high-affinity laminin receptor (LAMR), which is overexpressed in cancer cells. Sindbis virus is used as a gene therapy vector, however, it has exhibited oncolytic activity on cancer cells [149].

#### Sindbis Pre-Clinical Research

Sindbis virus has been shown to replicate and propagate in U87 glioma cell line in vitro and in vivo [92]. Sindbis has also been used as a vector expressing the gibbon ape leukemia virus membrane fusion glycoprotein gene (GALV.fus), which increases the infectivity of the virus and the cytotoxic effect in a U87 cell GBM mouse tumor model [150]. Sindbis vectors present a synergistic effect when combined with the chemotherapeutic agent paclitaxel [151] and the potential application for detection of the tumor cells in the brain parenchyma by the addition of reporter genes [152]. Future investigations are needed to elucidate the potential clinical use of this virus (Table 2).

### 3.8. Rift Valley Fever Virus (RVFV)

RFVF is a single stranded RNA virus that belongs to the *Bunyaviridae* family with a wide host-range, including several domestic animals and humans [153]. Although little is known about the clinical application of this virus in the field of brain tumors, there are some studies in the literature that prove its infection efficacy in vitro with the attenuated strains RVFV MP-12 and ZH548 in GBM C6 rat cells [154] and U87 [155].

## 4. Current OV Challenges for Malignant Glioma

The use of OVs is at the front edge of the next therapeutic approach in GBM treatment. The use of viruses, as with any other therapy, must deal with the stability and the specificity of the treatment without compromising the patients. The use of mammal adapted viruses requires modifications to limit viral replication and lytic effect on the tumor cells. However, viruses adapted to more distant species may be too attenuated to be effective and require some complementary modifications to boost specific toxicity against the tumor cells.

Important issues in the use of OV in patients remain unsolved, such as a complete lack of viral unspecific toxicity, DNA integration or viral latency in the host, virus restriction to the tumor cells, incomplete responses by attenuated viruses, and low capacity or complexity to be genetically modified. Productive viral infection to target brain tumors depends on several factors. One is the infiltrative growth pattern that requires optimal routes for vehicle delivery [116]. In this way, intratumoral administration is the most common and effective way to control concentration. Moreover, the blood brain barrier is an impediment even to small particles [158]. However, these invasive procedures are difficult and risky to make repeated doses [159]. To increase this therapeutic approach, combined synergistic therapies should be analyzed [160], such as the application of previous radiotherapy, the use of immunomodulators, or chemotherapeutics agents, which enhance cytotoxic effects [88,161]. It is also likely that a description of new viruses or modifications or the ones proposed thus far may improve current therapies.

A clearer definition of the tumor characteristics will determine future approximations to improve OV therapy. In recent years, an impressive effort to classify GBM genetic characteristics has uncovered tumor-specific modifications and key target points. However, much work is still needed to define the different scenarios of GBM immunocompetence and immunogenicity, in particular, those that will be involved in reacting to immunotherapeutic treatments and specifically the ones that use oncolytic viruses. The immunoprivileged characteristics of the CNS introduces an important uncertainty factor in generalizing the experiences of immune-based treatments to brain tumors. In addition, the immune characteristics of GBM cells require better animal models to understand the reactivity of different treatments. Some of the animal models used to test OV efficacy are immunocompromised, and thus the important contribution of the immune response to OV remains poorly characterized. Understanding the tumor microenvironment from this angle will determine the best strategy in each specific GBM case.

## 5. Conclusions

Based on current data from clinical trials, DNA viruses such as modified HSV-1 and adenovirus as well as RNA viruses such as reoviruses and NDV present promising results that require further improvements. Current versions of these viruses will need to be updated and tested in different glioma types and patient situations in order to improve glioma treatments. Additional viruses are still behind in proving some alternatives, but upcoming results from current clinical trials may provide various possibilities to improve the current OV therapeutic options.

Single treatments are unlikely to be enough in many tumors, including GBM. The combination of different treatments such as surgery, chemotherapy, radiotherapy, immunotherapy, and viral therapy will result in better treatments. Recent approaches propose the use of a sequential administration of immune-stimulating agonist to overcome tumor immune-tolerance. The introduction of therapeutic approaches based on trained immune stimulation may also provide a step forward in combinatorial therapies [162].

Finally, there is an urgent need to develop better immunocompetent animal models that consider different subtypes of GBM to better study and understand the best combination of treatments for these types of devastating tumors. In summary, OV is at the cutting edge of the next generation of glioma treatments and should be seriously considered as an option where no alternatives are available.

## Figures and Tables

**Figure 1 ijms-21-07604-f001:**
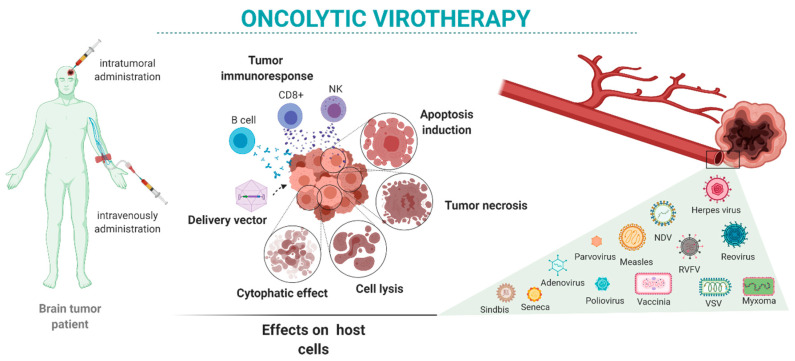
Oncolytic virotherapy in brain tumors: intratumor or systemic administration of oncolytic viruses may have different oncolytic reactivity once the virus reaches the target cells. Those effects depend on the characteristics of the tumor as well as the variety of available viruses and their characteristics.

**Table 1 ijms-21-07604-t001:** Preclinical studies of DNA viruses in glioma tumors.

Virus	Modifications	Cell Lines	In Vivo Models	Results
Herpes	*dls*ptk: TK deletion.	Human: U87 and T98G [22]	U87 i.c. and s.c. nude mice	Tumor cell infection and death. In vivo tumor reduction and increased surveillance.
	HSV-1716: γ_1_34.5 loci partial deletion.	Human: U87, T98G, SB18, U373 and U251 [24]	-	Tumor cell infection and death.
	G207: γ_1_34.5 loci deletion. ICP6 truncation.	Human: U87, U373, U138 and T98G [26]	U87 i.c. and s.c. nude mice, i.c. owl monkeys	Elimination of tumor cells, necrosis and no toxicity.
	rQNestin34.5: ICP6 deletion. γ_1_34.5 expression under Nestin promoter.	Human: U251, U87dEGFR, T98G, Gli36d5, U138, and MGH238 [28]	U87dEGFR i.c. and s.c. nude mice	Increase of oncolytic activity at in vitro and in vivo models
	NG34: γ_1_34.5 loci deletion. ICP6 deletion. GADD34 expression under Nestin promoter.	Human: U251, U87ΔEGFR and primary glioma cells Murine: GL261 [30]	U87ΔEGRF-RliFluc and G35 i.c. nude mice, BALB/c mice	Similar oncolytic activity as rQNestin34.5 with lower neurotoxicity.
	NG34scFvPD-1: γ_1_34.5 loci deletion. ICP6 deletion. GADD34 expression under Nestin promoter. scFvPD-1 expression under CMV’s IE promoter.	Human: U87ΔEGFR and U251 Murine: GL261N4 and CT2A [31]	GL261N4 and CT2A i.c. C57Bl/6J mice, GL261N4 and U87ΔEGFR i.c. nude mice	Increased oncolytic activity in comparison to NG34 in immunocompetent mice. Development of specific immunity and memory.
	G47Δ: γ_1_34.5 loci deletion. ICP6 truncation. α47 deletion. US11 expression under α47 promoter.	Human: U87 and U373 [34]	U87 s.c. in nude mice	Increased survival, higher number of cured mice than G207.
	C134: γ_1_34.5 loci deletion. HCMV’s IRS1 protein expression.	Human: D54, U87 and U251 Murine: N2A [32]	U87 i.c. in SCID mice	Reduced tumor volume and increased surveillance.
		Human and murine: 12 established GBM [33]	N2A orthotopic in A/J and BALB/c mice	Improved replication and longer survival *in vivo*
	HSV-1 R-LM113: insertion of scFvHER2 in gD protein.	Murine: established GBM [35]	PDGFB/DsRed-induced gliomas in nude mice	No toxicity in nude mice and oncolytic effect in HER2 overexpressing and established tumors in vivo.
	RAMBO: γ_1_34.5 loci deletion. ICP6 truncation. Vstat120 expression under IE4/5 HSV promoter.	Human: U343, U87, U87ΔEGFR, LN229, Gli36ΔEGFR-H2B-RFP, U251-T2, U87ΔEGFR-Luc [36]	U87ΔEGFR-Luc and Gli36ΔEGFR-H2B-RFP i.c. and U87ΔEGFR-Luc s.c. nude mice	Increased survival in vivo and inhibition of tumor vascularization.
	M002: γ_1_34.5 loci deletion. IL-12 expression.	Murine: 4C8 [37]	4C8 i.c. gliomas in B6D2F_1_ mice	Increase mice survival, infiltration of CD4+, CD8+ and NK cells. Longer viral persistence in tumors
	HSV-IL4: γ_1_34.5 loci deletion. IL-4 expression.	Human: U251 and D54 [21]	GL-261 i.c. in C57BL/6	Infiltration of macrophages, CD4+ and CD8+. Longer survival.
Adenovirus	ONIX-15: E1B-55kD deletion.	Human: 4 primary GBM [49]	S.c. xenograft in nude mice	Tumor regression.
	Delta-24-RGD: E1A partial deletion. RGD tripeptide incorporation.	Human: U251, U373, U87 and D54 [53]	D54 s.c. in nude mice	Cell death with low doses, single injection inhibits tumor growth, several injections resulted in 36% of animals with tumor regression.
	ICOVIR-5: E1A expression under E2F-1 promoter. E1A partial deletion. RGD tripeptide incorporation.	Human: U251 and U87 [54]	U87 i.c. xenograft in nude mice	Tumor cytotoxic effect in vitro high tumor selectivity and increase of survival in vivo.
Adenovirus	ICOVIR-17: E1A expression under a promoter including four palindromic E2F-1 sites and a Sp-1-binding site. E1A partial deletion. RGD tripeptide incorporation. PH20 expression under MLP promoter.	Human: U87, U138, LN308, Gli36, U373, LN229 and 6 primary GBM [60]	U87 and CSCs i.c. in nude mice	Better distribution in HA tumors. Longer mice survival.
	VCN-01: E1A expression under a promoter including four palindromic E2F-1 sites and a Sp-1-binding site. E1A partial deletion. RGD relocated in fiver shaft protein. PH20 expression under MLP promoter.	Human: U87, A172, T98G, U251, U373, SNB19 and 2 GBM CSC [61]	U87 and GBM CSC i.c. xenografts in nude mice	Control of tumor growth One single injection improves survival in aggressive infiltrative tumor.
	Delta-24-RGDOX: E1A partial deletion. RGD tripeptide incorporation. OX40L expression.	Human: U87 Murine: GL261 [62]	GL261 i.c. in C57BL/6 mice	Proliferation of tumor specific T cells. Sinergy with anti PD-L1.
	Delta-24-GREAT: E1A partial deletion. RGD tripeptide incorporation. GITRL expression.	Human: U87 and U251 Murine: GL261 [63]	GL261 i.c. in C57BL/6 mice	Extended survival and development of antiviral and antitumor specific response and memory.
	Ad-RTS-IL-12: No replicative. Expression of IL-12 under RTS^®^ system with veledimex as a co-treatment.	Murine: GL261 [64]	GL261 i.c. in C57BL/6 mice	Tumor infiltration with CD8, extended survival and immune memory development.
Vaccinia	rVV-p53: p53 expression.	Rat: C6 [69]	C6 s.c. in nude mice	Moderate cell apoptosis. Tumor growth control.
	rVV-mIL12/mIL2: IL12 expression. IL2 expression.	Rat: C6 [71]	C6 s.c. in nude mice	Cytokine toxicity at high dose Antitumor NK dependent effect.
	rVV-p53 and rVV-mL12: p53 expression. IL12 expression.	Rat: C6 [74]	C6 s.c. in nude mice	Better tumor growth control. Higher NK and macrophage infiltration.
	vvDD: TK deletion. VGF deletion.	Human: A172, U87MG and U118 Rat: RG2, F98 and C6 [73]	U87, U118 and C6 s.c. and RG2, F98 i.c. in nude mice	Control of tumor growth. Sinergy with rapamycin or cyclophosphamide.
			Rhesus macaques [75]	No adverse effects.
	vvDD-IL15Rα: TK deletion. VGF deletion. IL15Rα expression.	Murine: GL261 [75]	GL261 i.c. in C57BL/6J	Increase of NK and CD8+ in tumor. Prolonged survival.
	TG6002: TK deletion. ribonucleotide reductase genes deletion. FCU1 expression.	Human: U87 and patient derived GBM [76]	U87 i.c. and s.c. in nude mice	Prolonged survival in s.c. and i.c. Synergic effect with 5FC in i.c. model.
Myxoma	MYXV WT	Human: U87, U251, U373, U343, A172 and U118 Rat: RG2 and 9L [79]	U87 and U251 i.c. in nude mice	Regression and longer survival in both models.
		Human: U87, U251, and U118 [80]	U87 orthotopic in CB-17 SCID mice	Inhibition of MHC-I tumor expression and promotes NK mediated death.
		Human: U118 and 3 patient samples Murine: GL261 Rat: T9 [81]	-	SOC co-treatment increases results of MYXV.
	MYXV WT: administered in ADSCs	Human: U87 and U251 [82]	U87 orthotopic in nude mice	Increase the tumor infection rate
	MYXV-M011L-KO: M11L deletion	Human: Brain tumor initiating cells (BTIC) [80]	mBITCs i.c. in C57Bl/6J mice	Prolonged survival. TMZ increases oncolysis
Parvovirus		Human: U87 Rat: RG-2 [89]	U87 i-deficient rats and RG-2 i-competent	Complete remission of the tumors
	H-1PV WT	Human: U373, U138 and 5 CSCs [87] Human: U87 Rat: RG-2 [89]	RGD orthotopic ratsU87 i-deficient rats and RG-2 i-competent	Cathepsin B activation induces cell death in H-1PVComplete remission of the tumors
		Human: U87, U373, U118, MO59J and A172 Murine: GL261 [90] Human: U373, U138 and 5 CSCs [87]	U87 and U373 s.c. U87 orthotopic CB17-SCID miceRGD orthotopic rats	Selective infection, no toxicity, reduce tumor volume in vivo Cathepsin B activation induces cell death in H-1PV
	MVMp WT	Human: U373, U87, SW1088, SK-N-SH Rat: C6 [91] Human: U87, U373, U118, MO59J and A172 Murine: GL261 [90]	-U87 and U373 s.c. U87 orthotopic CB17-SCID mice	MVM p strain cytotoxic only in U373 and C6 (MVM) selective infection, no toxicity, reduce tumor volume in vivo
		Human: U87 and MO59J [92] Human: U373, U87, SW1088, SK-N-SH Rat: C6 [91]	-	Selective infection MVM p strain cytotoxic only in U373 and C6 (MVM)
		Murine: Fibroblast L929 and A9. Astrocytoma MT539MG [93], Human: U87 and MO59J [92]	-	Safe for microglia (MVMp) selective GBM infection (MVM)
		Murine: Fibroblast L929 and A9. Astrocytoma MT539MG [93],	-	Safe for microglia (MVMp)

i.c.: intracranial, s.c.: subcutaneous, i-deficient: immunodeficient, i-competent: immunocompetent; VGF: vaccinia growth factor; HCMV: human cytomegalovirus; TMZ: temozolomide; EGFR: epidermal growth factor receptor; MVM: murine minute virus.

**Table 2 ijms-21-07604-t002:** Preclinical studies of RNA viruses in glioma tumors.

Virus	Modifications	Cell Lines	In Vivo Models	Results
Reovirus	Reovirus	Human: 24 GBM cell lines [99]	U87 and U251 intracranial and subcutaneous in SCID mice	Death in 20 out of 24 GBM lines Regression in both in vivo models Toxicity in nude mice.
		Human: U87 and 2 patient-derived lines [100]	GL261 intracranial in C57/BL6	i.v. administration reaches brain tumors. T cell tumor recruitment and cytotoxicity. Synergy with anti PD-L1.
Measles	MV-CEA: CEA expression.	Human: U87, U251, and U118 [105]	U87 intracranial and subcutaneous in nude mice	Regression in s.c. tumor after intravenous and intratumor administration. Regression in intracranial tumor after intratumor administration.
	MV-NIS: NIS expression.	Human: U87, U251 and 6 patient derived GBM [106]	U251 subcutaneous and GBM intracranial in nude mice	Synergic effect of virotherapy and radiotherapy.
	MV-GFP-H_AA_-scEGFR: H protein partial deletion. scEGFR insertion in H protein.	Human: 5 patient derived GBM [107]	GBM intracranial in nude mice Mouse model Ifnar^ko^ CD46 Ge	Tumor regression after intratumor administration. No toxicity in CNS.
	MV-141.4: scFvCD133 insertion in H.	Human: primary GBM [111]	GBM intracranial in NOD/SCID	Better survival rate in comparison with MV-Edm.
VSV	VSV WT	-	C6 subcutaneous nude mice [112]	Inhibition of tumor growth.
	VSV-ΔG: G protein deletion.	Human: U87 Rat: C6 [111]	-	Infection of cell lines. Rapid lysis.
	VSV^ΔM51^: M51 single nucleotide deletion.	Human: 14 glioma cell lines and 15 primary gliomas [115]	U87 and U118 subcutaneous in nude mice	Infection and elimination of all cell lines. Tumor regression and prolonged survival.
	VSV-rp30: unknown viral glioma adaptation	Human: U87, U118, U373 and A172 [92]	U87 subcutaneous in nude mice	Increased selectivity and lytic capacity in glioblastoma cells. Tumor selectivity and cytopathic effect.
		Human: U87 and U118 [116]	U87 orthotopic in nude mice	Infection and lysis of brain and peripheral tumors.
	VSV-CT1/CT2: G protein partial deletion	Human: U87, U118, U373 and A172 [114]	U87 orthotopic in nude mice	Elimination of tumor cells. Normal cell toxicity can be eliminated with IFN co-treatment VSV-1p-GFP: infection and potent apoptosis over tumor.
	VSV-1p-GFP: GFP at the first position in the genome. VSV-CT9-M51: G protein partial deletion. **M51 single nucleotide deletion.**	Human: U87, U118, U373 and A172 Rat: 9L [119]	U87 orthotopic in CB17-SCID mice	VSV-CT1 and VSV-CT9-M51 have less toxicity than wt VSV. VSV-CT9-M51 is able to infect and kill tumors in brain.
	VSV-CT9-M51: G protein partial deletion. **M51 single nucleotide deletion.**	Human: primary GBM [122]	Orthotopic CB17 SCID	Coinfection with AAV-mIFN-β or with ribavirin enhances oncolytic properties.
Seneca Valley	SVV-001 WT	Human: primary GBM [138]	4 orthotopic models in nude mice	Partial response against glioma cells Effectivity in 2 of 4 in vivo tumors
		Human: GBM CSCs [139]	6 GBM CSC orthotopic nude mice	4 of 6 prolonged survival, tumor infection and cell lysis. Susceptibility dependent of sialic acid presence.
NDV	NDV WT	Human: 6 GBM CSCs [129]	Orthotopic nude mice	NDV replication is dependent on IFN deletion.
		Human: U87 and DBTRG.05MG [130]	Subcutaneous nude mice	Induce apoptosis. Decrease tumor volume.
		Human: A172 and U87 and 2 CSCs [131]	-	Induce apoptosis.
		Murine: GL261 [132]	GL261 orthotopic mice	NDV induces ICD.
		Human: T98G, LN18, U251, U87. Rat: C6 [133]	C6 in rats	Synergistic effects with TMZ. Decrease tumor volumes and increase OS.
Poliovirus	PVS-RIPO: IRES replaced with HRV2	Human: U87 [142]	-	Reduce viability.
		Human: CSCs and established cell lines [145]	HTB14 orthotopic and HTB15 flanks	Tumor regression.
		Human: 6 CSCs [143]	-	Cytolysis.
		Human: U87, HTB14 and HTB15 [144]	HTB15 in athymic Balb/c mice	Tumor regression.
Sindbis	Sindbis WT	Human: U87, U-118, U373, M059J, A172 [92]	U87 in flanks CB17-SCID mice	Effective replication and selective kill U87.
	Sindbis Gal.fu	Human: U87 [150]	U87 orthotopic in nude mice	Cytopathic activity.
RVFV	RVFV MP-12 and ZH548: attenuated strains	Rat: C6 [154] Human: U87 [155]	-	Infection occurs.

CEA: carcinoembryonic antigen; CNS: central nervous system; VSV: vesicular stomatitis virus; RVFV: RIFT valley fever virus; ICD: immunogenic cell death; OS: overall survival; PVS-RIPO: oncolytic polio/rhinovirus recombinant; IRES: internal ribosome entry site; HRV2: human rhinovirus 2.

**Table 3 ijms-21-07604-t003:** Clinical trials of oncolytic viruses (OVs) for glioma tumors.

Virus	Phase and Reference	*n* Patients	Results
Herpes	Phase I: HSV-1716 [38]	9	Two 24 moth survivors
	Phase Ib: HSV-1716 [39]	12	Evidence of tumor infection Three patients clinically stable for two years
	Phase II: HSV-1716 NCT02031965	2	No results available
	Phase I: G207 [41]	21	No toxicities
	Phase Ib: G207 [41]	6	No toxicity Evidence of tumor infection
	Phase I: G207 [42]	9	No toxicities in combination with 5 Gy
	Phase I: rQNestin34.5v2 NCT03152318	108	Recruiting
	Phase I: C134 NCT03657576	24	Recruiting
Adenovirus	Phase I: ONYX-015 [66]	24	No toxicity One patient without progression and some with regression
	Phase I: Delta-24-RGD NCT03896568	36	Recruiting
	Phase I: Delta-24-RGD NCT03178032	12	No results available
	Phase II: Delta-24-RGD NCT02798406	49	Active
	Phase I: Delta-24-RGD NCT02197169	37	No toxicities
	Phase I: Delta-24-RGD NCT01956734	31	No results available
	Phase I and II: Delta-24-RGD NCT01582516 [156]	20	Virus spread in tumor, oncolytic effect and immunostimulation
	Phase I: Delta-24-RGD NCT00805376	37	20% of >3 year survivors 12% of >95% tumor regression Evidence of immunostimulation
	Phase II Delta-24-RGD (2016-001600-40)	-	Discontinued
	Phase I: Delta-24-RGD NCT03714334	24	Recruiting
	Phase I: Delta-24-RGD NCT03072134	36	No results available
	Phase I: DNX-2440 NCT03714334	24	Recruiting
	Phase I/II: Ad-RTS-IL-12 NCT03330197	45	Recruiting
Reovirus	Phase I: Reovirus [101]	12	No toxicities
	Phase I: Reovirus NCT00528684 [102]	15	One 2 year survivor One 3 year survivor
	Phase Ib: Reovirus [100]	9	Evidence of T cell tumor infiltration and upregulation of IFN and PD-1/PD-L1 axis
	Phase I: Reovirus/Sargramostim NCT02444546	6	Active
Vaccinia	Phase I and II: TG6002 NCT03294486	78	Recruiting
Measles	Phase I: MV-CEA NCT00390299	23	No toxicities
NDV	Phase I/II: NDV-HUJ NCT01174537 [136]	14	No toxicities Complete regression in 1 patient
	Phase 0: MTH-68/H [134]	4	OS 5–9 years
	VOL-DC vaccine [135]	10	Increased OS
	Phase II: ATV-NDV vaccine [157]	23	PFS 40 weeks vs. 26 weeks
Parvovirus	H-1PV [94]	18	Enhanced immunogenicity
Poliovirus	Phase I: NCT01491893 [147]	61	No neurovirulence and increased survival rate
	Phase II: NCT02986178	122	Active
	Phase Ib: NCT03043391	12	Recruiting

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
