# Peer review of "Oncolytic Virotherapy in Glioma Tumors"

_ijms, 2020, doi:10.3390/ijms21207604_

Round 1

Reviewer 1 Report

Major comments

  • In this review Sergio et al. summarize the current status of OVs against glioma tumors. After a short introduction to the current situation of glioma treatment and a brief introduction on the OVs. The authors cited well the majority of OVs against glioma. However, the authors discussed other OVs either in preclinical or clinical studies. Many studies were discussed superficially; the connections or interpretations are mainly missing.
  • This review mainly discussed the genetic modification in detail rather than the impact of using the OV against glioma in detail. I recommended to make a table for all OVs and their genetic modification and focused more on the use of OVs against glioma either in preclinical or clinical.
  • All clinical trials are discussed without details. It is better to discuss some details and the progress, especially for the type of trial, it is monotherapy or OV- combined therapies, the OS survival, the problems in some trails and challenges.
  • All preclinical studies have been reported as follows “X virus can replicate in glioma cell lines and suppress the tumor growth” So, it would be difficult to get the efficacy differences between OVs. So it would be better to discuss the in-vivo models to test the oncolytic viruses against glioma, combination therapies, and the recent advances in the preclinical either through discussing the arming strategy or promising combination therapies.
  • This review mentioned the clinical trials and preclinical in one section. It would be better to separate the preclinical and clinical trials for each OV family.
  • It would be better to make a sub-heading for each oncolytic virus.

Minor comments:

  • The author stated that “OV has a double oncolytic action by both, directly destroying the cancer cells, sparing the patient’s life, and stimulating a tumor-specific immune response to revert the ability of tumors to escape the control of the immune system” what is the mean of, sparing the patient’s life in OVs mechanism?
  • The abstract need some clear explanation, especially for the aim and the construction of the review.
  • Line 90 “It has been demonstrated that this virus mediates a direct lytic effect in tumor cells, but in addition, most of the in vivo effects propose tumor destruction mediated by activation of tumor-specific immune responses” Does this mechanism specific for HSV viruses?
  • Line 95: A first approximation for oncolytic attenuation of HSV- 1 was HSV-1716” I think the first oHSV was dlspk done by Martuza in 1991. Please modify it.
  • Please add the following oHSVs ; rQNestin 34.5, Rambo, HSV- NG34 OV, HSV R-LM113 and NG34scFvPD-1 HSV viruses
  • Please add the armed OHSVs which were examined against glioma
  • The authors stated that no clinical trials or preclinical studies are using T-VEC. So what is the impact of mentioning T-VEC in glioma?
  • The genetic modification of some viruses and the impact of deletion or insertion of some genes such as G47 or ICOVIR-17 or ICOVIR-5 is difficult to understand.
  • The authors mentioned, “Until now, better approximations into the use of Adenovirus as oncolytic therapy are the Conditionally Replicative adenovirus (CRad)” why using CRad is better oncolytic adenovirus than other adenoviruses?
  • Please mention for the onyx-015 adenovirus approval in china.
  • Line 173, please correct the mistyping “this adenovirus that also includes de 24 base pair deletion of E1A”
  • Please mentioned for the genetic differences of ICOVIR-7 or ICOVIR-5 and ICOVIR-17.
  • Line 188 “immune stimulation is also another strategy that has been explored using adenoviruses to induce1 oncolysis and tumor regression. Delta-24-RGDOX is a modified Delta-24-RGD adenovirus that expresses the immune stimulatory OX40 ligand (OX40L)” Does Delta-24-RGDOX only the adenovirus express immune stimulatory genes?? There are others adeno-OVs were armed and tested in glioma. Please add them.
  • Please add information about Ad-RTS-hIL-12 with adenoviral vector.
  • Please add DNX-2401 OV- adenovirus
  • In vaccinia OV, mainly the authors discussed the armed oVVs. Please discuss how VV select the tumor cells and replicate.
  • Line 264: “One handicap of using MYXV as oncolytic virus in vivo is a low ability to proliferate outside the tumor injection area”. Do the author mean the effect on the contralateral side? This paper didn’t mentioned for the viral injection area.
  • Line 248: One of them is radiotherapy and TMZ pre-
    treatment followed by infection with MYXV, resulting in increased replication rate and decreased cell viability in GBM cell lines as compared to other non-tumor cells. Please discuss why radio or TMZ therapy increase the viral replication rate.
  • Line 267 “It has been proposed that H-1PV only destroy cisplatin and TRAIL resistant glioma cells by inducing cathepsins B and L aggregation and decreasing the expression of their inhibitors, the cystatin B and C” Does H-1PV destroy only the resistant glioma cells? How about the non-resistant cells?
  • Please discuss in details the First Phase I/IIa Glioblastoma Trial used Oncolytic H-1 Parvovirus (Ref 79)
  • Line 355; VSV-CT9 more toxic for no-tumor cells” what is the meaning of no-tumor cells?
  • Line 379 “Although the specific tropism for cancer cells is poorly understood” Do the authors mean the tumor cell tropism specific for NDV virus?
  • Line 426 “Sindbis vectors presents a synergistic effect when combined with chemotherapy” please define the type of chemotherapy.
  • Please discuss the attenuation strategy and tumor tropism for the following viruses MYXV, measles and reoviruses, Seneca valley,  Sindbis  RIFT Valley fever virus (RVFV)
  • The conclusion part seems to be challenges rather than summering the review article. Recommended to change it to challenges part and rewrite the conclusion part with considering the displayed information within the review.
  • In all tables. Please add separated column for references?
  • Please cite all tables in the context.

Author Response

October 6th, 2020

Dear Editor

Find included our reply to the Referees’ comments with respect to our review titled: “Oncolytic virotherapy in glioma tumors” (ijms-929846). We would like to thank the reviewers for their timely response and constructive suggestions. We believe that the addition of their comments has significantly improved the quality of the manuscript. In the following pages, we specify, point-by-point, the changes included in the new draft in line with Referees comments. We also indicate all the changes we have made in the manuscript with the track changes tool.

Reviewer #1

Major comments

 In this review Sergio et al. summarize the current status of OVs against glioma tumors. After a short introduction to the current situation of glioma treatment and a brief introduction on the OVs. The authors cited well the majority of OVs against glioma. However, the authors discussed other OVs either in preclinical or clinical studies. Many studies were discussed superficially; the connections or interpretations are mainly missing.

We agree with the Reviewer#1 that more explanation about the OVs in glioma could help to increase the impact of the review. For this reason, we have revised and amplified the information and connections throughout the manuscript.

This review mainly discussed the genetic modification in detail rather than the impact of using the OV against glioma in detail. I recommended to make a table for all OVs and their genetic modification and focused more on the use of OVs against glioma either in preclinical or clinical.

We have added a new column in the table with the OV genetic modifications in each assay.

All clinical trials are discussed without details. It is better to discuss some details and the progress, especially for the type of trial, it is monotherapy or OV- combined therapies, the OS survival, the problems in some trails and challenges.

We agree with the reviewer that a more detailed explanation of the clinical results will be very useful. We hope to have access to these data in the future, specially the one related to the immune response in patients receiving oncolytic viruses. A better description of the patients’ immune response to the GBM treatment both in the brain and systemically is needed.

Unfortunately, most of the clinical trials have not posted the results yet, however we have increased the information of the studies and added the following details in these paragraphs:

Phase I and Ib clinical studies in HSV-1 G207 demonstrated no neurovirulence in GBM patients even at high intracerebral inoculation doses (3x109 pfu) [40,41]. Moreover, its administration with radiotherapy in nine recurrent GBM patients showed a good response in six of them [42].

rQNestin34.5v2 virus is now under a Phase I clinical trial with 108 glioma patients in combination with cyclophosphamide, an immunomodulating drug that has shown an increase in tumor HSV viral titers and survival (NCT03152318) [43].

2.2.2 Adenovirus clinical studies

A phase I clinical trial using Onyx-015 was carried out with 24 glioma patients showing no adverse effects and regression in one patient and no progression of the disease in another participant [66]. Several phase I and II clinical trials have been carried out using both Delta-24 virus and the Delta-24 RGD version with promising results. Delta-24-RGDOX is now in a Phase I clinical trial with 24 GBM patients (NCT03714334). Intratumoral administration of Ad-RTS-hIL-12 is now under a Phase I/II clinical trial in patients with pediatric brain tumors in combination with oral administration of veledimix (NCT03330197) (Table 3).

2.3.2 VV clinical studies

Despite the high number of preclinical studies in VV, confirming the safety and oncolytic activity, it has not yet transferred into successful clinical trials and only TG6002 is being evaluated. The main study consists of a phase I clinical trial with 78 GBM patients using a combination of TG6002 and 5-FC. 24 patients of this preliminary study without tumor progression will be included in the next phase IIa study (NCT03294486) (Table 3).

2.5.2 Parvoviridae clinical studies

Following this promising data, a phase I/IIa clinical trial was conducted in 18 unifocal recurrent GBM patients. Participants were divided into two groups. First, one of the groups was treated intravenously and the other group received an intratumoral injection. The second H-1PV administration (into the tumor cavity during surgery) was the same in both groups (ParvOryx01: NCT01301430) [94]. As an initial approach the safety and the tolerance were evaluated, indicating a lack of toxicity and that upon intravenous administration, the virus was able to cross the blood-brain barrier and reach the tumor [95]. Additionally, H-1PV was able to enhance immunogenicity within the tumor microenvironment [96]. H-1PV treated patients displayed an increase in tumor-infiltrating cytotoxic T cells, and induction of cathepsin B, and the expression of IFN-γ and IL-2, among other cytokines within the tumor microenvironment (Table 3).

  3.2.1 Reovirus clinical studies

Single reovirus intratumoral administration in 12 recurrent glioma showed no adverse effects in a Phase I clinical trial [101]. Similar observations were obtained using an intratumoral infusion of 72 h in 18 participants [102]. Recently, a Phase Ib clinical trial posted an increase in tumor leukocyte infiltration and higher expression of IFN, caspase 3 and PD-L1 in tumors from reovirus treated patients [100] (Table 3).

3.2.2 Measles clinical studies

A phase I clinical trial using MV-CEA has been carried out in 23 recurrent participants. One group was treated directly into the resectioned cavity and the other patients were also treated before the surgery by a catheter. They observed a tolerance up to 107 pfu TCID50 and no significant differences between groups in progression free survival (PFS) at 6 months (NCT00390299) (Table 3).

One of the main limitations in using MV in brain tumor patients is the invasive intratumoral administration procedure due to the low efficacy when the virus is administered intravenously [109].

3.4.2 Poliovirus clinical studies

Although some studies have been done in small cohorts of GBM patients, all of them have shown promising results. Csatary and colleagues treated intravenously 4 GBM patients with the mesogenic strain MTH-68/H. They reported an increase in the survival rates to 5-9 years together with an enhancement of in the quality of life, after conventional treatment [134]. Similar effects were observed in 10 patients treated with the vaccine VOL-DC composed of NDV infected dendritic cells [135]. One additional report describes that repeated intravenous administration of the lentogenic NDV strain OV001/HUJ in 14 GBM patients achieved a complete tumor regression in one patient [136] (Table 3).

All preclinical studies have been reported as follows “X virus can replicate in glioma cell lines and suppress the tumor growth” So, it would be difficult to get the efficacy differences between OVs. So it would be better to discuss the in-vivo models to test the oncolytic viruses against glioma, combination therapies, and the recent advances in the preclinical either through discussing the arming strategy or promising combination therapies.

We appreciate the suggestion. We have introduced the following paragraphs:

Line 180-182: The pre-clinical use of these genetically modified viruses has demonstrated an increase in HSV tumoral selectivity and an enhancement in some immune evasion gene expression. For these reasons, modified HSVs have been assessed for glioma treatment in clinical studies.

Line 210-212 Until now, the best approximations into the use of replicative Adenovirus as oncolytic therapyhave been the Conditionally Replicative adenovirus (CRad). Different generations of CRad have been developed in recent years with promising results in gliomas in both pre-clinical and clinical studies [46].

 Line 342-344: Despite the high number of preclinical studies in VV, confirming the safety and oncolytic activity, it has not yet transferred into successful clinical trials and only TG6002 is being evaluated. The main study consists of a phase I clinical trial…

Line 376-377: These findings together with further information from ongoing studies have a potential to make MYXV a viable option for brain tumor clinical management.

Line 404-405: The notable safety and tolerability in systemic and local administration makes Parvoviridae a good viral family candidate to treat GBM combined with other immunotherapeutic agents.

Line 430-431: Despite these preliminary safety and tolerability results, there are currently no available clinical trials assessing the impact in the patient's survival.

 Line 472-474: One of the main limitations in using MV in brain tumor patients is the invasive intratumoral administration procedure due to the low efficacy when the virus is administered intravenously [109].

Line 535-537: Most of the VSV experiments were done in immunocompromised mouse models, so future assays assessing the impact in the systemic immune response should be developed to translate these clinical research outcomes into clinical practice.

This review mentioned the clinical trials and preclinical in one section. It would be better to separate the preclinical and clinical trials for each OV family.

Following his/her advice, and to clarify the review, we have made two sections in each OV family: the preclinical research (Table 1 and 2) and the clinical trials (Table 3).

It would be better to make a sub-heading for each oncolytic virus.

We really appreciate this suggestion; we have now divided each OV family with subheadings.

Minor comments:

The author stated that “OV has a double oncolytic action by both, directly destroying the cancer cells, sparing the patient’s life, and stimulating a tumor-specific immune response to revert the ability of tumors to escape the control of the immune system” what is the mean of, sparing the patient’s life in OVs mechanism?

We thank the reviewer to highlight it. We have deleted this confusing sentence.

The abstract needs some clear explanation, especially for the aim and the construction of the review.

We agree with the Reviewer #1. We have added some new information to the abstract as follows:

“Here, we review the state-of-the-art oncolytic virotherapies for the treatment of brain tumors (Figure 1). We present the main viruses proposed for brain tumor oncolytic therapy alone and in combination with other therapeutic approaches. We focus on viruses used in preclinical studies (Table 1 and 2) and clinical trials (Table 3) performed mostly in GBM patients.”

Line 90 “It has been demonstrated that this virus mediates a direct lytic effect in tumor cells, but in addition, most of the in vivo effects propose tumor destruction mediated by activation of tumor-specific immune responses” Does this mechanism specific for HSV viruses?

Thank you for raising this point.

While this effect has been shown by different oncolytic viruses in the context of glioma (ex. rAd Delta-24-RGDOX line 198), lack of in vivo experiments using immunocompetent mice makes it difficult to interpret immunostimulatory effect of most of the proposed oncolytic viruses. As we indicate at several places in the manuscript, immune-competent orthotopic GBM animal tumor models are needed to better understand tumor reactivity to these type of therapies.

Line 95: A first approximation for oncolytic attenuation of HSV- 1 was HSV-1716” I think the first oHSV was dlspk done by Martuza in 1991. Please modify it.

We thank you for providing this data. We have modified this sentence at line 95 by adding dlsptk HSV virus.

Please add the following oHSVs ; rQNestin 34.5, Rambo, HSV- NG34 OV, HSV R-LM113 and NG34scFvPD-1 HSV viruses

We really appreciate this suggestion. These oHSVs have been added in the HSV section.

Please add the armed OHSVs which were examined against glioma

We understand for armed OV all the viruses that have been genetically modified to increase their oncolytic effect. The major amount of HSVs included in this manuscript belong to this category. 

Bibliography:

Passaro, C.; Alayo, Q.; De Laura, I.; McNulty, J.; Grauwet, K.; Ito, H.; Bhaskaran, V.; Mineo, M.; Lawler, S.E.; Shah, K.; Speranza, M.C.; Goins, W.; McLaughlin, E.; Fernandez, S.; Reardon, D.A.; Freeman, G.J.; Chiocca, E.A.; Nakashima, H. Arming an Oncolytic Herpes Simplex Virus Type 1 with a Single-chain Fragment  Variable Antibody against PD-1 for Experimental Glioblastoma Therapy. Clin. cancer Res.  an Off. J. Am. Assoc.  Cancer Res. 2019, 25, 290–299, doi:10.1158/1078-0432.CCR-18-2311.

The authors stated that no clinical trials or preclinical studies are using T-VEC. So what is the impact of mentioning T-VEC in glioma?

This virus was added for being approved for melanoma treatment and its great results in the treatment of different types of cancer in order to present it as a possible candidate for future glioma treatment assays. Following your recommendation has been removed from the manuscript.

The genetic modification of some viruses and the impact of deletion or insertion of some genes such as G47 or ICOVIR-17 or ICOVIR-5 is difficult to understand.

We appreciate Reviewer #1 for this comment. Genetic modifications of these viruses have been rewritten.

The authors mentioned, “Until now, better approximations into the use of Adenovirus as oncolytic therapy are the Conditionally Replicative adenovirus (CRad)” why using CRad is better oncolytic adenovirus than other adenoviruses?

Thanks for the comment. Although oncolytic effect of both types of viruses may depend on the particular patient characteristics, a general estimation of viral toxicity suggest that tumor restricted replication will be less noxious to the patient. Conditionally Replicative adenovirus, or CRad, are replicative Adenoviruses that are restricted to grow in tumor cells. Lower toxicity elicited by this virus due to inefficacy infecting normal cells makes them in opinion safer in the use as oncotherapy.

Please mention for the onyx-015 adenovirus approval in china.

We want to thank Reviewer #1 for this suggestion. ONIX-015 approval for head and neck cancer treatment in China has been added at line 215 and in the Table 1.

Line 173, please correct the mistyping “this adenovirus that also includes de 24 base pair deletion of E1A”

We thank Reviewer #1  for letting us know about this mistake. It has been corrected.

Please mentioned for the genetic differences of ICOVIR-7 or ICOVIR-5 and ICOVIR-17.

We really appreciate this suggestion. We have included the genetic modifications of these viruses.

Line 188 “immune stimulation is also another strategy that has been explored using adenoviruses to induce1 oncolysis and tumor regression. Delta-24-RGDOX is a modified Delta-24-RGD adenovirus that expresses the immune stimulatory OX40 ligand (OX40L)” Does Delta-24-RGDOX only the adenovirus express immune stimulatory genes?? There are others adeno-OVs were armed and tested in glioma. Please add them.

We thank Reviewer #1 for the comment. We have added Ad-RTS-hIL-12 and Delta-24-GREAT as other adenovirus based OV aimed to elicit immune stimulation.

Please add information about Ad-RTS-hIL-12 with adenoviral vector.

We appreciate Reviewer #1 for this comment. Ad-RTS-hIL-12 and Ad-RTS-mIL-12 have been added to the manuscript.

Please add DNX-2401 OV- adenovirus

We really appreciate the Reviewer #1 comment. DNX-2401 is an alternative name for Delta-24-RGD virus that has been described in the text. We have added this alternative name next to the original at line 501.

Bibliography:

Martínez-Vélez, N.; Garcia-Moure, M.; Marigil, M.; González-Huarriz, M.; Puigdelloses, M.; Gallego Pérez-Larraya, J.; Zalacaín, M.; Marrodán, L.; Varela-Guruceaga, M.; Laspidea, V.; Aristu, J.J.; Ramos, L.I.; Tejada-Solís, S.; Díez-Valle, R.; Jones, C.; Mackay, A.; Martínez-Climent, J.A.; García-Barchino, M.J.; Raabe, E.; Monje, M.; Becher, O.J.; Junier, M.P.; El-Habr, E.A.; Chneiweiss, H.; Aldave, G.; Jiang, H.; Fueyo, J.; Patiño-García, A.; Gomez-Manzano, C.; Alonso, M.M. The oncolytic virus Delta-24-RGD elicits an antitumor effect in pediatric glioma and  DIPG mouse models. Nat. Commun. 2019, 10, 2235, doi:10.1038/s41467-019-10043-0.

In vaccinia OV, mainly the authors discussed the armed oVVs. Please discuss how VV select the tumor cells and replicate.

We appreciate this comment. VV tropism and replication cycle characteristics have been added.

Line 264: “One handicap of using MYXV as oncolytic virus in vivo is a low ability to proliferate outside the tumor injection area”. Do the author mean the effect on the contralateral side? This paper didn’t mentioned for the viral injection area.

MYXVs are not able to spread and colonize the whole tumor after intratumoral injection, so its effectiveness as OV is restricted by this characteristic. 

Line 248: One of them is radiotherapy and TMZ pre-
treatment followed by infection with MYXV, resulting in increased replication rate and decreased cell viability in GBM cell lines as compared to other non-tumor cells. Please discuss why radio or TMZ therapy increase the viral replication rate.

Thank you for the comment, indeed chemotherapy or radiotherapy don´t change the MYXV replication rate, it increases the spread of infection. We have corrected in the text.

Line 267 “It has been proposed that H-1PV only destroy cisplatin and TRAIL resistant glioma cells by inducing cathepsins B and L aggregation and decreasing the expression of their inhibitors, the cystatin B and C” Does H-1PV destroy only the resistant glioma cells? How about the non-resistant cells?

In the study we cited, they investigated the effect of H-1PV in glioma cells that have acquired resistance to these drugs. However, it has been well established that H-1PV kill other glioma cells.

We have now modified it on the text.

Bibliography:

Di Piazza, M.; Mader, C.; Geletneky, K.; Herrero y Calle, M.; Weber, E.; Schlehofer, J.; Deleu, L.; Rommelaere, J. Cytosolic Activation of Cathepsins Mediates Parvovirus H-1-Induced Killing of Cisplatin and TRAIL-Resistant Glioma Cells. J. Virol. 2007, 81, 4186–4198, doi:10.1128/jvi.02601-06.

Geletneky, K.; Hartkopf, A.D.; Krempien, R.; Rommelaere, J.; Schlehofer, J.R. Improved Killing of Human High-Grade Glioma Cells by Combining Ionizing Radiation with Oncolytic Parvovirus H-1 Infection. J. Biomed. Biotechnol. 2010, 2010, 350748, doi:10.1155/2010/350748.

Please discuss in details the First Phase I/IIa Glioblastoma Trial used Oncolytic H-1 Parvovirus (Ref 79)

We have added the following information:

“Following this promising data, a phase I/IIa clinical trial was conducted in 18 unifocal recurrent GBM patients. Participants were divided into two groups. First, one of the groups was treated intravenously and the other group received an intratumoral injection. The second H-1PV administration (into the tumor cavity during surgery) was the same in both groups (ParvOryx01: NCT01301430) [94].” 

Line 355; VSV-CT9 more toxic for no-tumor cells” what is the meaning of no-tumor cells?

We refer to normal cells not forming part of any tumor. Sentence has been rephrased.

Line 379 “Although the specific tropism for cancer cells is poorly understood” Do the authors mean the tumor cell tropism specific for NDV virus?

We thank the Reviewer for bringing out this important topic in the OV field.

As the Reviewer knows some viruses have a specific tropism for tumoral cells, and others need a genetic modification to acquire it.

In this case, NDV is able to replicate in cancer cells with some alterations in antiviral or apoptotic pathways, as for example the IFN signaling route.

Bibliography:

Zamarin, D.; Palese, P. Oncolytic Newcastle disease virus for cancer therapy: old challenges and new directions. Future Microbiol. 2012, 7, 347–367, doi:10.2217/fmb.12.4.

Badrinath, N.; Heo, J.; Yoo, S.Y. Viruses as nanomedicine for cancer. Int. J. Nanomedicine 2016, 11, 4835–4847, doi:10.2147/IJN.S116447.

Line 426 “Sindbis vectors presents a synergistic effect when combined with chemotherapy” please define the type of chemotherapy.

The chemotherapeutic agent used in this study was paclitaxel which binds to the microtubules and blocks cell cycle.

We have added this information in the main text.

References:

Horwitz, S.B. Taxol (paclitaxel): mechanisms of action. Ann. Oncol.  Off. J. Eur. Soc. Med. Oncol. 1994, 5 Suppl 6, S3-6.

Please discuss the attenuation strategy and tumor tropism for the following viruses MYXV, measles and reoviruses, Seneca valley,  Sindbis  RIFT Valley fever virus (RVFV)

Thank you for noting this. Indeed, MYXV-M011L-KO and Measles virus genetic modifications have been already explained in the text.

There are different strategies to increase viruses oncolytic activity, for example:

-In reovirus it has been published some attenuation strategies such as TRDSA-, VeroAV, Jin-1, Jin-2, Jin-3, Y354H, among others. However, any of them have been evaluated in gliomas.

-The same occurs with Sindbis virus, in which some modifications (for example: AR339 strain) have been reported in cervical, ovarian and oral squamous cancer cells, but not in gliomas.

-RVFV is considered a bioterrorism agent, so attenuated strains that allow it to work in a BSL-2 have been studied. In the case of gliomas, the variants RVFV MP-12 or the ZH548 strains were analyzed.

Finally, to the best of our knowledge, no modifications have been developed in Seneca valley virus.

We have added some information about cell tropism in the main text.

Bibliography:

Mohamed, A.; Johnston, R.N.; Shmulevitz, M. Potential for Improving Potency and Specificity of Reovirus Oncolysis with Next-Generation Reovirus Variants. Viruses 2015, 7, 6251–6278, doi:10.3390/v7122936.

Unno, Y.; Shino, Y.; Kondo, F.; Igarashi, N.; Wang, G.; Shimura, R.; Yamaguchi, T.; Asano, T.; Saisho, H.; Sekiya, S.; Shirasawa, H. Oncolytic Viral Therapy for Cervical and Ovarian Cancer Cells by Sindbis Virus AR339 Strain. Clin. Cancer Res. 2005, 11, 4553–4560.

Saito, K.; Uzawa, K.; Kasamatsu, A.; Shinozuka, K.; Sakuma, K.; Yamatoji, M.; Shiiba, M.; Shino, Y.; Shirasawa, H.; Tanzawa, H. Oncolytic activity of Sindbis virus in human oral squamous carcinoma cells. Br. J. Cancer 2009, 101, 684–690, doi:10.1038/sj.bjc.6605209.

Harmon, B.; Schudel, B.R.; Maar, D.; Kozina, C.; Ikegami, T.; Tseng, C.-T.K.; Negrete, O.A. Rift Valley Fever Virus Strain MP-12 Enters Mammalian Host Cells via Caveola-Mediated Endocytosis. J. Virol. 2012, 86, 12954–12970, doi:10.1128/jvi.02242-12.

Ritter, M.; Bouloy, M.; Vialat, P.; Janzen, C.; Haller, O.; Frese, M. Resistance to Rift Valley fever virus in Rattus norvegicus: Genetic variability within certain “inbred” strains. J. Gen. Virol. 2000, 81, 2683–2688, doi:10.1099/0022-1317-81-11-2683.

Burke, M.J. Oncolytic Seneca Valley Virus: past perspectives and future directions. Oncolytic virotherapy 2016, 5, 81–89, doi:10.2147/OV.S96915.

The conclusion part seems to be challenges rather than summering the review article. Recommended to change it to challenges part and rewrite the conclusion part with considering the displayed information within the review.

We added a new section called “Current OV challenges for malignant glioma” and changed the conclusion section according to this comment.

In all tables. Please add separated column for references?

We have followed the bibliography style of the journal, although if the Reviewer #1 still considers that a new column should be included in the table, we will be willing to follow his directions.

Please cite all tables in the context.

We have now clarified this in the text.

Reviewer 2 Report

The authors of this review paper describe the main oncolytic virus-based therapies proposed as therapeutic means to destroy glioma cells in vitro and to trigger tumor destruction in vivo and deeply revisit many of the more prevalent preclinical and clinical data for DNA-and RNA-viruses. In addition, they provide an overview of some key features of oncolytic virus-based therapies in gliomas in general.

In their introduction, the authors shortly summarize the demografics and outcomes of glioma and glioblastoma patients. Of note, TTF is approved as first-line treatment according to the EF-16 trial (and not as a therapy in relapse), and this should be set into context. In addition, a deeper insight in the biology of glioma with a focus on reasons why glioblastomas are difficult to treat by immuno-targeted therapies as oncolytic viruses should be provided. Some of these reasons are the location of gliomas behind the blood brain barrier, a low activity of the immune system in the brain, an immunosuppressive environment and then observation that oncolytic viruses are not always selective for tumor cells and do not uniformly infect all tumor cells.

In the main section, which is the strong part of the manuscript, the authors provide an extensive review of several types of viruses, sorted into DNA and RNA viruses. This part provides a good overview over most of the viruses that were / are used in preclinical and clinical studies sin glioma so far. However, I would suggest to put a little more emphasis on the clinical parts of these developments, to increase translational aspects that may be the most relevant for clinician scientists reading this review.

The weaker part of the manuscript is that the authors to not discuss in sufficient depth why oncolytic viruses, even after decades of development, have not been translated into clinical practice. The authors themselves name some of these reasons, e.g. viral unspecific toxicity, lack of DNA integration or viral latency in the host, lacking virus restriction to the tumor cells, incomplete responses by attenuated viruses and a low capacity to genetically modify the virus adequately. They however leave out other possible reasons of failure. Some of these reasons are inherent to drug delivery aspects, the infiltrative behavior of gliomas, the fact that not all tumor cells are infected even in target regions of the oncolytic virus, the lack of scientifically well-developed combination therapies and the lack of well-designed randomized trials with adequate controls. It would also be interesting to hear about aspects how to overcome these problems.

Minor points are that some of the revisited literature is not adequately cited. E.g., if the authors cite clinical data as e.g. on page 11, line 415, they should cite the full data including overall survival data for the experimental and standard arms of the trial. In addition, nominators of given numbers are sometimes missing as e.g. on page 10, line 387. Please revise the paper throughout for leave-outs like that. Lastly, the English language should be substantially improved, possibly by the help of a native speaker.

Author Response

Reviewer #2

In their introduction, the authors shortly summarize the demografics and outcomes of glioma and glioblastoma patients. Of note, TTF is approved as first-line treatment according to the EF-16 trial (and not as a therapy in relapse), and this should be set into context. In addition, a deeper insight in the biology of glioma with a focus on reasons why glioblastomas are difficult to treat by immuno-targeted therapies as oncolytic viruses should be provided. Some of these reasons are the location of gliomas behind the blood brain barrier, a low activity of the immune system in the brain, an immunosuppressive environment and then observation that oncolytic viruses are not always selective for tumor cells and do not uniformly infect all tumor cells.

We really appreciate this comment. We have made the necessary changes in the text and updated the bibliography contents in the Introduction section with the recent results of the Randomized Clinical Trial and added a new paragraph explaining the OVs challenges in the brain tumor field.

Line 44-44: “However, new radiotherapy regimens such as tumor treating fields (TTFields) have increased the overall survival by some months. Despite these aggressive therapies, unfortunately most of the tumors relapse and the majority of GBM patients die within 21 months [5].

Line 75-77: “There are many hurdles to overcome for the successful clinical application of OVs, such as the immunosuppressive GBM microenvironment or the presence of the blood brain barrier. In fact, the high number of infiltrating immune cells and the cancer stem cells impairs selective viral replication [17] . 

References:

Chinot, O.L.; De La Motte Rouge, T.; Moore, N.; Zeaiter, A.; Das, A.; Phillips, H.; Modrusan, Z.; Cloughesy, T. AVAglio: Phase 3 trial of bevacizumab plus temozolomide and radiotherapy in newly diagnosed glioblastoma multiforme. Adv. Ther. 2011, 28, 334–340, doi:10.1007/s12325-011-0007-3.

Martikainen, M.; Essand, M. Virus-based immunotherapy of glioblastoma. Cancers (Basel). 2019, 11, doi:10.3390/cancers11020186.

In the main section, which is the strong part of the manuscript, the authors provide an extensive review of several types of viruses, sorted into DNA and RNA viruses. This part provides a good overview over most of the viruses that were / are used in preclinical and clinical studies sin glioma so far. However, I would suggest to put a little more emphasis on the clinical parts of these developments, to increase translational aspects that may be the most relevant for clinician scientists reading this review.

We appreciate this observation from Reviewer #2, which is also a concern for Reviewer #1. For this reason, we have added more detailed information about the clinical trials including the number of patients involved and some of the most relevant results. However, our main goal is to summarize the current state of OVs opportunities for targeted therapeutics, more than focus on the detailed results.

The weaker part of the manuscript is that the authors to not discuss in sufficient depth why oncolytic viruses, even after decades of development, have not been translated into clinical practice. The authors themselves name some of these reasons, e.g. viral unspecific toxicity, lack of DNA integration or viral latency in the host, lacking virus restriction to the tumor cells, incomplete responses by attenuated viruses and a low capacity to genetically modify the virus adequately. They however leave out other possible reasons of failure. Some of these reasons are inherent to drug delivery aspects, the infiltrative behavior of gliomas, the fact that not all tumor cells are infected even in target regions of the oncolytic virus, the lack of scientifically well-developed combination therapies and the lack of well-designed randomized trials with adequate controls. It would also be interesting to hear about aspects how to overcome these problems.

We thank the Reviewer #2 for highlighting these important topics in the field of oncolytic virotherapy. Following her/his advice we have added this new paragraph in the section 5. Current OV challenges for malignant glioma:

 Productive viral infection to target brain tumors depends on several factors. One is the infiltrative growth pattern that requires optimal routes for vehicle delivery [116]. In this way, intratumoral administration is the most common and effective way to control concentration. Moreover, the blood brain barrier is an impediment even to small particles [158]. However, these invasive procedures are difficult and risky to make repeated doses [159]. To increase this therapeutic approach, combined synergistic therapies should be analyzed [160] such as the application of previous radiotherapy, the use of immunomodulators or chemotherapeutics agents which enhance cytotoxic effects [88,161].It is also likely that a description of new viruses or modifications or the ones proposed so far, may improve current therapies. 

Bibliography:

Ozduman, K.; Wollmann, G.; Piepmeier, J.M.; van den Pol, A.N. Systemic vesicular stomatitis virus selectively destroys multifocal glioma and  metastatic carcinoma in brain. J. Neurosci. 2008, 28, 1882–1893, doi:10.1523/JNEUROSCI.4905-07.2008.

Li, L.; Liu, S.; Han, D.; Tang, B.; Ma, J. Delivery and Biosafety of Oncolytic Virotherapy. Front. Oncol. 2020, 10, 1–15, doi:10.3389/fonc.2020.00475.

Howells, A.; Marelli, G.; Lemoine, N.R.; Wang, Y. Oncolytic viruses-interaction of virus and tumor cells in the battle to eliminate cancer. Front. Oncol. 2017, 7, doi:10.3389/fonc.2017.00195.

Minor points are that some of the revisited literature is not adequately cited. E.g., if the authors cite clinical data as e.g. on page 11, line 415, they should cite the full data including overall survival data for the experimental and standard arms of the trial. 

We have added that information in the line 593-598:

“Therefore, an interventional clinical study (NCT01491893) with 61 recurrent GBM patients was developed to confirm that the intratumoral injection of the virus is safe and increases the survival rate to 21% at 36 months whereas this was 4% in the historical control group [147]. With the aim of confirming the safety and test efficacy of the virus, there are currently two active clinical trials for recurrent glioma. An interventional phase II study with 122 enrolled adult patients (NCT02986178) and a Phase Ib with 12 malignant glioma children (NCT03043391) are ongoing (Table 3).”

We have also reviewed the complete manuscript and added more information about the clinical settings.

Line 394-397: “They reported an increase in the survival rates to 5-9 years together with an enhancement of in the quality of life, after conventional treatment [134].”

In addition, nominators of given numbers are sometimes missing as e.g. on page 10, line 387. Please revise the paper throughout for leave-outs like that.

We have made the necessary arrangements in the main text to correct these mistakes.

Lastly, the English language should be substantially improved, possibly by the help of a native speaker.

We have reviewed and corrected the English language and the grammatical typos throughout the manuscript. The text has been corrected and edited by a scientific English language editor Brian Crilly.

Round 2

Reviewer 1 Report

I think the manuscript has been significantly improved and now I accepted for publication of the manuscript in IJMS

Author Response

We appreciate Reviewer´s comments. They have be very useful to improve the manuscript.

Reviewer 2 Report

The authors have further developed the manuscript and have addressed all concerns of the reviewer. In the new version, the manuscript is more informative for translational scientists and physicians alike. However, some concerns, predominantly formal ones, remain: 

They should again formally revise the manuscript and erase formatting and typo issues. Spaces in between lines and text blocks should be homogenous throughout the manuscript.

On page 14, lines 462 ff., the citation should be formatted in a proper way. 

Abreviations in headings and subheadings should be avoided, e.g. "VV" on page 8, lines 336.

Author Response

Reviewer 2:

The authors have further developed the manuscript and have addressed all concerns of the reviewer. In the new version, the manuscript is more informative for translational scientists and physicians alike. However, some concerns, predominantly formal ones, remain: 

We appreciate reviewer´s comments and we believe that their critiques have helped to improve the manuscript.

They should again formally revise the manuscript and erase formatting and typo issues. Spaces in between lines and text blocks should be homogenous throughout the manuscript.

We have extensively read the manuscript trying to remove the typos as indicated. In addition, we have now corrected the spaces between paragraphs to have an uniform text.

On page 14, lines 462 ff., the citation should be formatted in a proper way. 

We apologize, we are not sure if author is referring to reference 106 or other particular reference. We believe that reference 106 is well cited but we may be missing some point. We appreciate if the reviewer can help us in the correction by pointing out the specific reference we need to modify.

Abbreviations in headings and subheadings should be avoided, e.g. "VV" on page 8, lines 336.

Thanks for noticing. The text has been changed, in line 377 of page 8:

“2.3.2. Vaccinia virus clinical studies”

Editors:

in the final text, the Authors  should be named as NAME SURNAME, not the opposite order. Otherwise readers may be mistaken and the litterature indexing platforms as well. E.g., reviewer 1 thought that the paper was from Sergio et al., instead it is from Rius-Rocabert et al. This is a pretty naive mistake from the Authors, that must be absolutely corrected.

Thank you for the comment, we apologize. The text has been corrected in lines 21 and 22 accordingly.

21 “Sergio Rius-Rocabert 1,2,3,†, Noemí García-Romero 4,†, Antonia García 3, Angel Ayuso-Sacido 22 4,5,* and Estanislao Nistal-Villan 1,2,*”

Round 3

Reviewer 2 Report

All suggested changes have been made. Thank you to the authors for following our suggestions.